# *Amos*: An Adam-style Optimizer with Adaptive Weight Decay towards Model-Oriented Scale

## Abstract

We present *Amos*, a stochastic gradient-based optimizer designed for training deep neural networks. It can be viewed as an Adam optimizer with theoretically supported, adaptive learning-rate decay and weight decay. A key insight behind *Amos* is that it leverages model-specific information to determine the initial learning-rate and decaying schedules. When used for pre-training BERT variants and T5, Amos consistently converges faster than the state-of-the-art settings of AdamW, achieving better validation loss within $\leq 70\%$ training steps and time, while requiring $\leq 51\%$ memory for slot variables. Our code is open-sourced at: `https://anonymous-url`.

## 1 Introduction

The Adam (Kingma & Ba, 2015) optimizer is widely used for training deep neural networks, demonstrating fast convergence especially in the early stages of training. Although previous works have found issues regarding the theoretical convergence of Adam as the training proceeds (Reddi et al., 2018), in practice it is remedied by various learning-rate schedules and weight decay (Loshchilov & Hutter, 2019). Specifically, Adam with linear learning-rate decay and constant weight decay is the standard setting for pre-training large language models such as BERT (Devlin et al., 2019). However, these decay settings are usually ad-hoc, increase the number of hyper-parameters, and may introduce additional complexities in usage. For example, the linearly decaying learning-rate schedule requires knowing the number of training steps in advance, which makes it nontrivial to continuously train a model after the learning-rate decays to $0$.

In this work, we present *Amos*, a new optimizer with a theoretically supported and adaptive schedule for learning-rate and weight decay, which can significantly outperform the state-of-the-art AdamW settings for pre-training language models, provide guidance for hyper-parameter tuning, reduce the memory usage, and train continuously without having to specify the number of training steps a priori.

A key insight behind Amos is a hyper-parameter $\tilde{\eta}$ to be provided by the *model architecture*, which indicates the expected scale of the trainable weights $\tilde{\theta}$ of the model (§ 2), i.e., theoretically we assume that an optimal point $\tilde{\theta}^*$ exists within the $|\tilde{\theta}^*| \leq \tilde{\eta}$ diameter. Deep neural networks are likely to satisfy such a constraint without degrading performance, because there exist many good local minima; and we show that an appropriate $\tilde{\eta}$ for Amos can improve generalization and accelerate convergence (§ 5.2). In this work, $\tilde{\eta}$ is calculated in a consistent way from the input/output scale of neural network components, which is hinted by the model design (§ A.4). Given $\tilde{\eta}$, Amos decides a learning-rate per variable, and its L2 regularization will lead the trained weights to the specified scale. The decay of the learning-rate is then determined by the L2 regularizer. Thus, Amos performs better because it can utilize the model-oriented information $\tilde{\eta}$ efficiently; the name *Amos* stands for "**A**daptive weight decay towards **M**odel-**O**riented **S**cale".

Empirically, we focus on the Transformer architecture (Vaswani et al., 2017) since it is pre-dominant in pre-trained language models (Bommasani et al., 2021), but add additional experiments on LSTM (Gers et al., 2000) and ResNet (He et al., 2016). We apply Amos to the pre-training of 4 models: BERT (Devlin et al., 2019), two Transformer variants with relative position representations (Su et al., 2021; Shaw et al., 2018), and the T5 model (Raffel et al., 2020); some with various model sizes and batch sizes. In all experiments, Amos consistently outperforms the state-of-the-art setting, achieving better validation loss within $\leq 70\%$ training steps and time (§ 5.1). Compared to AdamW,

the memory usage for slot variables is reduced to $\leq 51\%$ in Amos (§ A.8). In addition, Amos does not calculate learning-rate from a maximum number of training steps, so one can seamlessly continue training from any checkpoints, which is not trivial for AdamW with linear learning-rate decay (§ 5.1).

## 2 THE ALGORITHM

For notation, we denote model weights by $\tilde{\boldsymbol{\theta}}$, and an online learning algorithm recursively calculates a sequence of weights, $\tilde{\boldsymbol{\theta}}_1, \tilde{\boldsymbol{\theta}}_2, \ldots$, from initial weights $\tilde{\boldsymbol{\theta}}_0$ and training examples $z_t$ at each step $t = 0, 1, \ldots$. An optimizer uses the gradient $\tilde{\boldsymbol{g}}_t = \nabla \ell(z_t; \tilde{\boldsymbol{\theta}}_t)$ to compute a weight update $\tilde{\boldsymbol{\theta}}_{t+1} \leftarrow \tilde{\boldsymbol{\theta}}_t - \tilde{\boldsymbol{\delta}}_t$, in order to minimize the loss function $\ell(z; \tilde{\boldsymbol{\theta}})$. In neural network models, the model weights $\tilde{\boldsymbol{\theta}}$ is an array of trainable tensors (i.e. variables) collected from all model components; we view a variable and its slices as subsets of the model weights (e.g. $\boldsymbol{\theta} \subseteq \tilde{\boldsymbol{\theta}}$ is a variable slice that functions in part of the model). We use a bold letter to denote an array (e.g. $\boldsymbol{\theta}_t$, $\tilde{\boldsymbol{\theta}}_t$), and the same normal letter to denote a scalar element of that array (e.g. $\theta_t$) for describing element-wise operations. We use tilde for information of the whole model (e.g. $\tilde{\boldsymbol{\theta}}_t$), and drop tilde to indicate subsets (e.g. $\boldsymbol{\theta}_t$).

To start, we recall the update rule of the RMSProp optimizer (Tieleman & Hinton, 2012), which computes the weight update by $\delta_t \leftarrow \frac{\alpha}{\sqrt{v_t}} g_t$, where $\alpha$ is a scalar learning-rate and $v_t$ a running average of the squared gradients $g_t^2$. Based on this, Adam (Kingma & Ba, 2015) replaces $g_t$ with its running average $m_t$ (i.e. momentum), and adopts bias correction $\hat{m}_t$, $\hat{v}_t$ for running averages. Further, AdamW (Loshchilov & Hutter, 2019) allows a schedule for learning-rate $\alpha_t$ (depending on the step $t$) and adds a weight decay: $\delta_t \leftarrow \alpha_t \left( \frac{1}{\sqrt{\hat{v}_t}} \hat{m}_t + \gamma \theta_t \right)$, where $\gamma$ is a constant hyper-parameter. For pre-training Transformer variants, the learning-rate schedule $\alpha_t$ is set to linearly decay to 0 after warm-up. Therefore, a maximum number of training steps before the learning-rate decays to 0 has to be set as a hyper-parameter. Amos, with a similar construction, has the following update rule:

$$\boldsymbol{\delta}_t \leftarrow d_t \left( \frac{\xi \eta}{\sqrt{\hat{v}_t}} \boldsymbol{g}_t + \frac{1}{2} \gamma_t \boldsymbol{\theta}_t \right) \quad \text{where} \quad \gamma_t \leftarrow c_t \frac{\xi^2}{\hat{v}_t} \mathrm{M}_2(\boldsymbol{g}_t)^2. \tag{1}$$

Here, $\mathrm{M}_2(\boldsymbol{a}) := \sqrt{\frac{1}{k} \sum_{i=1}^{k} a_i^2}$ denotes the quadratic mean of entries of an array $\boldsymbol{a} \in \mathbb{R}^k$. The update rule consists of a gradient descent part (the term containing $\boldsymbol{g}_t$) and an L2 regularization part (the term containing $\boldsymbol{\theta}_t$)[1], similar to AdamW. The full Amos is shown in Algorithm 1. We explain several novel aspects below.

*Model-oriented scale*: For each variable $\boldsymbol{a} \subseteq \tilde{\boldsymbol{\theta}}$ in the model weights, we specify the scale $\eta(\boldsymbol{a})$ we expect $\boldsymbol{a}$ to converge to, i.e. $\mathrm{M}_2(\boldsymbol{a}^*) \approx \eta$ for an optimal $\tilde{\boldsymbol{\theta}}^* \supseteq \boldsymbol{a}^*$. Different variables may have different scale $\eta$'s. For a common case of a linear transformation, $\boldsymbol{y} = \boldsymbol{x}\boldsymbol{W} + \boldsymbol{u}$ ($\boldsymbol{W}, \boldsymbol{u} \subseteq \tilde{\boldsymbol{\theta}}$, $\boldsymbol{W} \in \mathbb{R}^{m \times n}$, $\boldsymbol{x} \in \mathbb{R}^m$), we calculate $\eta(\boldsymbol{W})$ by assuming that $\boldsymbol{x}$ is random Gaussian with standard deviation $\sigma_x$, and $\boldsymbol{y}$ random Gaussian with standard deviation $\sigma_y$; so we have $\eta(\boldsymbol{W}) = \sigma_y / (\sigma_x \sqrt{m})$ in order to satisfy the input/output standard deviation (assuming entries of $\boldsymbol{W}$ to be Gaussian as well). Additionally, we set $\eta(\boldsymbol{u}) = \sigma_y / 2$ to ensure that $\boldsymbol{u}$ has a slightly smaller magnitude than $\boldsymbol{x}\boldsymbol{W}$. The input/output standard deviation can be hinted by other layers in the model; for example, the activation function GELU (Hendrycks & Gimpel, 2016) usually expects the inputs to have standard deviation $\approx 1$, because its non-linearity mostly lies within that range; also the output standard deviation of LayerNormalization (Ba et al., 2016) is expected to be 1. For Transformer variants, we will discuss the input/output standard deviation of all types of non-linear layers and derive $\tilde{\boldsymbol{\eta}}$ in § A.4.

*Factored initial learning-rate:* In Amos, we use $\xi \eta$ as the initial learning-rate, where $\eta$ is the model-oriented scale specified for each variable, and $\xi$ is a global learning-rate shared across all variables. For online optimizers, the learning-rate is generally affected by both data and model; by factoring the initial learning-rate into $\xi$ and $\eta$, we disentangle the two to some extent: While $\xi$ is tuned and may depend on the data, $\tilde{\boldsymbol{\eta}}$ is calculated from the model architecture.

---

[1]Following Loshchilov & Hutter (2019), we decouple the gradient of an L2 regularization term (taking the form of a weight decay) apart from the adaptive gradient normalization factor $\frac{1}{\sqrt{\hat{v}_t}}$. When an adaptive optimizer is used, Loshchilov & Hutter (2019) point out that the decoupled weight decay is not equivalent to the L2 regularization without explicit decoupling, and the former is more appropriate. In this work, we always treat L2 regularization as decoupled weight decay, and use the two terms interchangeably.

*Adaptive L2 regularization*: Unlike AdamW which uses a constant $\gamma$ for weight decay, the Amos weight decay $\tilde{\gamma}_t$ is intended to control the scale of trained variables, rather than regularize the loss function; so $\gamma_t$ decays to 0 at $t \to \infty$ to be less biased, and it is adaptive in the sense that $\tilde{\gamma}_t$ depends on $\tilde{g}_t$ so that the variables not getting gradient updates are not regularized. Thus, the L2 regularization is robust to sparse gradients, and it does not introduce any additional hyper-parameter. We will give a heuristic derivation of the form of $\gamma_t$ in § 4.

*Decay factors*: $\tilde{d}_t$, $\tilde{c}_t$ are per-parameter decay factors such that $d_0 = c_0 = 1$ and $d_t$, $c_t$ monotonically decrease to 0 at $t \to \infty$. We provide a theoretical derivation of the asymptotic behavior of these factors in § A.2, together with a default form that works well empirically in all our experiments. The decay factors do not depend on a maximum number of training steps, thus enabling arbitrary continuous training.

*Memory Reduction*: Most previous optimizers operate element-wise, so the slot variables (e.g. the running average $\tilde{v}_t$, $\tilde{m}_t$ in Adam) have the same shape as $\tilde{\theta}$, which can be memory consuming. In Amos, two slot variables ($\tilde{v}_t$, $\tilde{b}_t$ in Algorithm 1) are shared by certain slices in the model weights, reducing the memory usage of these slot variables. For example, if $\mathbb{R}^{m \times n} \ni W \subseteq \tilde{\theta}$ is a linear transformation, the corresponding $v_t \in \mathbb{R}^{1 \times n}$ is shared by the input dimension of $W$, reducing the memory usage by $m$ times. As a result, in Equation 1 and Algorithm 1, $v_t$, $b_t$, $c_t$ and $d_t$ are reduced and become scalars, to be used and updated by vector-valued $g_t$ and $\theta_t$. In this work, we reduce the input dimension of linear transformations, the embed dimension of embedding matrix, and all dimensions for other variables by default. An ablative study with different settings is found in § A.8.

---

**Algorithm 1** The *Amos* optimizer at step $t$.

---

**Input** $\tilde{g}_t = \nabla \ell(z_t; \tilde{\theta}_t)$: The gradient of loss $\ell$ on a random example $z_t$.

**Input** $\tilde{\theta}_t$: Trainable model weights at step $t$.

**Input** $\tilde{v}_{t-1}, \tilde{b}_t$: Slot variables of shape broadcastable to $\tilde{\theta}$, initialized to $\mathbf{0}$.

**Input** (Optional) $\tilde{m}_t$: Slot variable of the same shape as $\tilde{\theta}$, initialized to $\mathbf{0}$ for momentum.

**Hyper-parameter** $\xi$: Global learning-rate.

**Hyper-parameter** $\tilde{\eta}$: Expected scale for model weights $\tilde{\theta}$.

**Hyper-parameter** $\tilde{c}_t$: Decay factor for L2 regularization. Defaults to $c_t = \left(1 + \frac{1}{4}\sqrt{\xi}b_t\right)^{-\frac{1}{2}}$.

**Hyper-parameter** $\tilde{d}_t$: Decay factor for learning-rate. Defaults to $d_t = \left(1 + \frac{1}{4}\sqrt{\xi\eta}b_t\right)^{-1}$.

**Hyper-parameter** $\beta \in [0, 1)$: Exponential decay rate for running average $\tilde{v}_t$.

1: (Optional) $g_t \leftarrow \dfrac{\chi}{\max(\chi, |g_t|)} g_t$          ▷ Gradient clipping with hyper-parameter $\chi > 0$.

2: $v_t \leftarrow \beta v_{t-1} + (1 - \beta)\,\mathrm{M}_2(\boldsymbol{g}_t)^2$          ▷ Running average of squared gradients.

3: $\hat{v}_t \leftarrow v_t / (1 - \beta^t)$          ▷ Bias correction.

4: $\gamma_t \leftarrow c_t \dfrac{\xi^2}{\hat{v}_t}\,\mathrm{M}_2(\boldsymbol{g}_t)^2$          ▷ Adaptive L2 regularization strength.

5: $\boldsymbol{\delta}_t \leftarrow d_t \left( \dfrac{\xi\eta}{\sqrt{\hat{v}_t}} \boldsymbol{g}_t + \dfrac{1}{2}\gamma_t \boldsymbol{\theta}_t \right)$          ▷ Amos update rule.

6: $b_{t+1} \leftarrow b_t + \gamma_t(1 + b_t)$          ▷ Decay factor update.

7: (Optional) $\delta_t \leftarrow m_{t+1} \leftarrow \mu m_t + (1 - \mu)\delta_t$          ▷ Momentum with hyper-parameter $\mu \in [0, 1)$.

**Output:** Updated model weights $\theta_{t+1} \leftarrow \theta_t - \delta_t$.

**Output:** Updated slot variables $\tilde{r}_t$, $\tilde{b}_{t+1}$ and optional $\tilde{m}_{t+1}$.

---

**Hyper-parameter Tuning**    The running average $v_t$ in Amos is a low-cost estimator for $\mathbb{E}[\mathrm{M}_2(\boldsymbol{g}_t)^2]$, where the expectation is taken over the example $z_t$ randomly drawn from the training data. It is similar to $v_t$ in Adam except that the mean square $\mathrm{M}_2(\boldsymbol{g}_t)^2$ is used instead of element-wise $g_t^2$, due to the memory reduction. Thus, the hyper-parameter $\beta$ behaves similarly to $\beta_2$ in Adam: Since the estimator mostly depends on the previous $1/(1 - \beta)$ steps, $\beta$ should be close enough to 1 to make the estimator accurate, but not too large that the model weights in the previous $1/(1 - \beta)$ steps differ too much from the current step. We set $\beta = 0.999$ by default (the same as $\beta_2$ in Adam), and it is found that $\beta$ should be smaller with larger batch size (Shazeer & Stern, 2018; Liu et al., 2019).

The global learning-rate $\xi$ can depend on the step $t$ to follow a warm-up schedule at the beginning of training, but a schedule with learning-rate decay is not necessary, since the decay factor $\tilde{d}_t$ is already included in Algorithm 1; most of the time $\xi$ remains a constant. While this constant is the major hyper-parameter to be tuned, a good rule of thumb is to set $\xi$ to the same order of magnitude as $1/\sqrt{N}$, where $N$ is the number of independent batches in the training set (see § 4 for a justification). This value is usually larger than the typical learning-rates used for Adam. It also implies that $\xi$ should be in proportion to the square-root of the batch size, which we observe in practice as well (§ A.5).

In addition, Algorithm 1 includes optional gradient clipping and momentum. Momentum in Amos is applied *after* the main update rule (unlike Adam which applies it before). It can improve performance for pre-training Transformer variants, but consume memory because the slot variable $\tilde{m}_t$ must have the same shape as $\tilde{\theta}$. When momentum is applied, its decay rate $\mu$ is typically set to $0.9$.

## 3 RELATED WORK

Besides RMSProp (Tieleman & Hinton, 2012), Adam (Kingma & Ba, 2015) and AdamW (Loshchilov & Hutter, 2019), the number of previous works on optimization is vast, so we focus on some directly related alternatives below. Also, we note that Amos is a stochastic first-order optimizer, in contrast to recent progress in the second-order optimization methods (Gupta et al., 2018). The convergence of stochastic optimizers has been studied in terms of stochastic approximation (Bottou, 1998), regret (Hazan, 2019), or nonconvex stochastic programming (Ghadimi & Lan, 2013). In particular, Reddi et al. (2018) observed cases of non-convergence of Adam (with constant learning-rate) and proposed a fix. In our work, we analyze the behavior of Amos in an intuitive and heuristic manner, but leave a rigorous convergence proof (e.g. based on regret) to future work.

**AdaGrad:** The update rule of AdaGrad (Duchi et al., 2011) is $\delta_t \leftarrow \frac{\alpha}{\sqrt{b_t}} g_t$, where $b_{t+1} \leftarrow b_t + g_t^2$ is similar to the $b_t$ in Algorithm 1, in the sense that both AdaGrad and Amos use a (weighted) sum of squared gradients to decay learning-rates. Such decay is "adaptive" because the learning-rate will decay more for parameters getting more updates, which is suitable for sparse gradients. On the other hand, conventional wisdom is that the learning-rate in AdaGrad might decay "too fast" in some cases, which makes the convergence slow, and Adam mitigates this issue by using a running average of squared gradients instead of the decay factor. However, the AdamW setting suggests that the normalization factor by running average of squared gradients is not a replacement for learning-rate decay; one still needs a linearly decaying learning-rate schedule for better convergence. Thus, Amos integrates both the Adam-style gradient normalization and the AdaGrad-style learning-rate decay; with gradient normalization, the learning-rate can actually decay faster and it converges faster.

**SGD with L2 Regularization:** For the classic Stochastic Gradient Descent (SGD) algorithm, it is recommended to decay the learning-rate by the factor $\frac{1}{\lambda t}$, where $\lambda$ is the smallest eigen-value of the Hessian (Murata, 1998). Although $\lambda$ is generally unknown, adopting an L2 regularizer of strength $\lambda'$ guarantees that $\lambda \geq \lambda'$, so one can set the learning-rate to $\frac{1}{\lambda' t}$ (Bottou, 2012). In Amos, we adopt a similar idea to heuristically derive the learning-rate decay (see § A.3 for more detailed discussion), by connecting the decaying speed with the strength of L2 regularization (i.e., the L2 strength $\gamma_t$ in Algorithm 1 also appears in the update of $b_t$). Unlike SGD, both the learning-rate and L2 regularization in Amos decay adaptively. The adaptive L2 regularization, in particular, is a novel component unseen in previous optimizers.

**LAMB:** The LAMB optimizer (You et al., 2020) and its origin LARS (You et al., 2017) share several similar aspects with Amos. The idea of layer-wise learning-rate in LAMB and LARS is similar to the per-variable learning-rate $\tilde{\eta}$ in Amos; they all normalize the gradients in some way; and they all imply scaling up the learning-rate as the batch size increases. In our experiments, scaling the global learning-rate of Amos in proportion to the square-root of the batch size indeed works (§ A.5), although we leave a systematic study of scaling-up to extremely large batch sizes and comparing with LAMB and LARS to future work.

**AdaFactor:** In Adam, the slot variable $\tilde{v}_t$ for maintaining running average of squared gradients requires the same amount of memory as the model weights $\tilde{\theta}$. In order to reduce the memory usage, AdaFactor (Shazeer & Stern, 2018) proposes to use nonnegative matrix factorization to decompose any matrix into two vectors. In contrast, Amos reduces memory usage by simply reducing some axes

of the slot variables and broadcasting to the shape of model weights. This reduction is more efficient than AdaFactor, and our experiments suggest that it will not degrade performance (§ A.8).

## 4 DERIVATION OF AMOS

In this section, we heuristically derive the Amos update rule (Equation 1). We start from a general form of the weight update for a given variable $\boldsymbol{\theta}$,

$$\boldsymbol{\theta}_{t+1} = \boldsymbol{\theta}_t - \alpha_t \boldsymbol{g}_t \quad \text{where} \quad \tilde{\boldsymbol{g}}_t = \nabla\ell(z_t; \tilde{\boldsymbol{\theta}}_t), \tag{2}$$

and gradually pin down to the specific form of Equation 1. Here, the step size $\alpha_t > 0$ is a scalar (due to our memory reduction mechanism in § 2) and is shared across the elements of the vector-valued $\boldsymbol{g}_t, \boldsymbol{\theta}_t \in \mathbb{R}^k$. We are focusing on a subset of model parameters, but furthermore note that $\alpha_t$ may differ for different variables.

Then, the following Descent Lemma (Murata, 1998) provides a sanity check for a wide range of possible forms of $\alpha_t$, while also suggests some constraints. Its proof can be found in § A.1.

**Lemma 4.1** (Descent Lemma). *If $\alpha_t$ does not depend on $z_t$, then there exists $\epsilon_t > 0$ such that*

$$\mathbb{E}_t[\mathbb{E}_{t+1}[\ell(z_{t+1}; \tilde{\boldsymbol{\theta}}_{t+1})]] \leq \mathbb{E}_t[\ell(z_t; \tilde{\boldsymbol{\theta}}_t)] \quad \text{for any} \ \alpha_t < \epsilon_t,$$

*where $\mathbb{E}_t[\bullet]$ denotes the expectation taken over the random example $z_t$ drawn from the training data at step $t$, while conditioned on $z_{t-1}, \ldots, z_0$ of the previous steps.*

In light of Lemma 4.1, we require (I) $\alpha_t$ *does not depend on $z_t$ (but may differ for different variables)*, and (II) $\alpha_t$ *decays to 0 at $t \to \infty$*, so the step-size can be sufficiently small that the Descent Lemma applies and Equation 2 will always make progress on average.

In the Amos update rule, $\alpha_t = d_t \frac{\xi\eta}{\sqrt{\hat{v}_t}}$ and $\hat{v}_t$ depends on $z_t$, which seems to violate requirement (I) above. However, $\hat{v}_t$ should be regarded as an approximation of $\mathbb{E}[\mathrm{M}_2(\boldsymbol{g}_t)^2]$, where $\mathbb{E}[\bullet]$ denotes the expectation taken over examples randomly drawn from the training data, which is $z_t$ independent.

Next, we add an L2-regularization term to Equation 2:

$$\boldsymbol{\theta}_{t+1} = \boldsymbol{\theta}_t - (\alpha_t \boldsymbol{g}_t + \rho_t \boldsymbol{\theta}_t) \tag{3}$$

where $\rho_t \geq 0$ can depend on $\boldsymbol{g}_t$ (hence "adaptive"), but we require (III) $\mathbb{E}[\rho_t]$ *does not depend on $\boldsymbol{g}_t$*. The intuition behind is that an L2-regularization should have the same strength across all variables, rather than be affected by the typical gradient magnitude on each variable. It is the same intuition that motivates the weight decay decoupled from gradient adaptive factors (Loshchilov & Hutter, 2019).

The first challenge for Amos is to keep a balance between $\alpha_t$ and $\rho_t$, so that $\mathrm{M}_2(\boldsymbol{\theta}_t)$ will converge to the pre-specified, per-variable hyper-parameter $\eta$. In order to achieve this, we will declare some intuitions on the largeness of $\boldsymbol{g}_t$, $\mathbb{E}[\boldsymbol{g}_t]$ and $\rho_t \boldsymbol{\theta}_t$, as a guide for our heuristic derivation. For deep neural networks, $\boldsymbol{g}_t$'s upon different $z_t$'s appear to be randomly noisy, so *they will cancel out* when being averaged to $\mathbb{E}[\boldsymbol{g}_t]$; which means that $\mathrm{M}_2(\mathbb{E}[\boldsymbol{g}_t])$ is usually much smaller than $\mathrm{M}_2(\boldsymbol{g}_t)$. On the other hand, $\boldsymbol{\theta}_t$ does not depend on $z_t$, and it changes slowly between different steps, so the update by $\rho_t \boldsymbol{\theta}_t$ is easier to accumulate than $\alpha_t \boldsymbol{g}_t$. This means that the magnitude of $\rho_t \boldsymbol{\theta}_t$ can be kept smaller than $\alpha_t \boldsymbol{g}_t$ while still compete with $\alpha_t \mathbb{E}[\boldsymbol{g}_t]$. In Amos, $\rho_t = d_t \frac{1}{2} c_t \frac{\xi^2}{\hat{v}_t} \mathrm{M}_2(\boldsymbol{g}_t)^2$ decays to 0 faster than $\alpha_t$ (due to the extra decay factor $c_t$), which we assume will make $\rho_t \boldsymbol{\theta}_t$ small enough compared to $\alpha_t \boldsymbol{g}_t$, when $t$ is large.

Quantitatively, we consider the error $\tilde{\boldsymbol{\varepsilon}}_t = \tilde{\boldsymbol{\theta}}_t - \tilde{\boldsymbol{\theta}}^*$, where $\tilde{\boldsymbol{\theta}}^*$ is a local minimum. Equation 3 implies

$$\mathrm{M}_2(\boldsymbol{\varepsilon}_{t+1})^2 = \mathrm{M}_2(\boldsymbol{\varepsilon}_t)^2 - \frac{2}{k}(\alpha_t \boldsymbol{g}_t + \rho_t \boldsymbol{\theta}_t) \cdot \boldsymbol{\varepsilon}_t + \mathrm{M}_2(\alpha_t \boldsymbol{g}_t + \rho_t \boldsymbol{\theta}_t)^2$$

$$\approx \mathrm{M}_2(\boldsymbol{\varepsilon}_t)^2 - \frac{2}{k}(\alpha_t \boldsymbol{g}_t + \rho_t \boldsymbol{\theta}_t) \cdot \boldsymbol{\varepsilon}_t + \alpha_t^2 \mathrm{M}_2(\boldsymbol{g}_t)^2, \tag{4}$$

where we investigate a time point $t$ large enough that the model nearly converges. At this point, $\rho_t \boldsymbol{\theta}_t$ is small compared to $\alpha_t \boldsymbol{g}_t$, so it can be approximately omitted in the third term. And we should have

$\mathbb{E}[\boldsymbol{g}_t] \approx \boldsymbol{0}$ and $\mathrm{M}_2(\boldsymbol{\theta}_t) \approx \eta$ if the trained weights converge to scale $\eta$. So taking $\mathbb{E}[\bullet]$ of Equation 4, we should get

$$\mathbb{E}[\mathrm{M}_2(\boldsymbol{\varepsilon}_{t+1})^2] \approx \mathrm{M}_2(\boldsymbol{\varepsilon}_t)^2 - \frac{2}{k}\mathbb{E}[\rho_t]\boldsymbol{\theta}_t \cdot \boldsymbol{\varepsilon}_t + \alpha_t^2\mathbb{E}[\mathrm{M}_2(\boldsymbol{g}_t)^2].$$

Furthermore, in order for the model to converge, we should have $\mathbb{E}[\mathrm{M}_2(\boldsymbol{\varepsilon}_{t+1})^2] \leq \mathrm{M}_2(\boldsymbol{\varepsilon}_t)^2$ from the above. Hence, we should have

$$\alpha_t^2\mathbb{E}[\mathrm{M}_2(\boldsymbol{g}_t)^2] \leq \frac{2}{k}\mathbb{E}[\rho_t]\boldsymbol{\theta}_t \cdot \boldsymbol{\varepsilon}_t \leq 2\mathbb{E}[\rho_t]\,\mathrm{M}_2(\boldsymbol{\theta}_t)\,\mathrm{M}_2(\boldsymbol{\varepsilon}_t) \approx 2\mathbb{E}[\rho_t]\eta\,\mathrm{M}_2(\boldsymbol{\varepsilon}_t)$$

as a necessary condition for the trained weights to converge to scale $\eta$. By setting $\rho_t$ to the smallest possible, we get

$$2\rho_t\eta\,\mathrm{M}_2(\boldsymbol{\varepsilon}_t) = \alpha_t^2\,\mathrm{M}_2(\boldsymbol{g}_t)^2, \tag{5}$$

which is an important relation connecting $\rho_t$ to $\alpha_t$. We require *(IV) Equation 5 to be satisfied throughout the course of training*, and use it ubiquitously in our derivation. It is out of the scope of this work to prove whether Equation 5 actually makes $\mathrm{M}_2(\boldsymbol{\theta}_t)$ converge to $\eta$; but the requirements so far already determine a basic form of the Amos update rule (as shown in Lemma 4.2 below), and our experiments suggest that Amos indeed brings the trained weights to the specific scale (§ 5.2).

**Lemma 4.2** (Basic Form of Amos). *Assume Equation 5, requiring that $\alpha_t$ does not depend on $z_t$ and $\mathbb{E}[\rho_t]$ does not depend on $\boldsymbol{g}_t$. Then, we have*

$$\alpha_t \propto \frac{1}{\sqrt{\mathbb{E}[\mathrm{M}_2(\boldsymbol{g}_t)^2]}} \quad \text{and} \quad \rho_t \propto \frac{\mathrm{M}_2(\boldsymbol{g}_t)^2}{\mathbb{E}[\mathrm{M}_2(\boldsymbol{g}_t)^2]}.$$

The proof is found in § A.1. It is noteworthy that the Adam-style gradient normalization naturally occurs in $\alpha_t$. Based on Lemma 4.2, Amos is derived by specifying the initial learning-rate and decay schedule. For that, we need the following assumption to quantify the largeness of $\boldsymbol{g}_t$ and $\mathbb{E}[\boldsymbol{g}_t]$.

**Assumption 1.** A scalar $\xi > 0$ exists such that $\dfrac{\mathrm{M}_2(\mathbb{E}[\boldsymbol{g}_t])}{\sqrt{\mathbb{E}[\mathrm{M}_2(\boldsymbol{g}_t)^2]}} \geq \xi$ for all $t$ and across all variables.

This assumption formalizes two intuitions, i.e. randomly noisy $\boldsymbol{g}_t$ will cancel out when being averaged to $\mathbb{E}[\boldsymbol{g}_t]$ (so $\xi$ has a small value), and as the training proceeds, $\mathrm{M}_2(\boldsymbol{g}_t)$ will decrease[2] along with $\mathrm{M}_2(\mathbb{E}[\boldsymbol{g}_t])$ (so the ratio remains larger than a constant $\xi > 0$). Assumption 1 is verified by our experiments (§ A.6).

The value of $\xi$ is related to the global learning-rate in Amos (as shown in Lemma 4.3 below), which is tuned as a hyper-parameter in practice. However, we also provide an intuitive estimation of $\xi$, which is usually a good start for hyper-parameter tuning. The intuition is to view the canceling out of $\boldsymbol{g}_t$ averaged to $\mathbb{E}[\boldsymbol{g}_t]$ as similar to the average of $N$ i.i.d. samples drawn from a distribution of mean $\boldsymbol{0}$. According to the Law of Large Numbers, the variance of the average (i.e. $\mathrm{M}_2(\mathbb{E}[\boldsymbol{g}_t])^2$) is about $1/N$ of the variance of the distribution (i.e. $\mathbb{E}[\mathrm{M}_2(\boldsymbol{g}_t)^2]$), so $\xi \approx \frac{1}{\sqrt{N}}$. In reality, the gradients of deep neural networks, computed over mini-batches, appear to be highly random. So $N$ is usually of the same order of magnitude as the number of independent batches in the training data.

Now, we can derive the optimal initial learning-rate as below, under an ideal condition that $\boldsymbol{g}_0$ points to the same direction as $\boldsymbol{\varepsilon}_0$. The proof is found in § A.1.

**Lemma 4.3** (Initial Learning-rate). *Assume Equation 2, Assumption 1, $\alpha_0 = \alpha/\sqrt{\mathbb{E}[\mathrm{M}_2(\boldsymbol{g}_0)^2]}$ and that $\boldsymbol{g}_0$ points to the same direction as $\boldsymbol{\varepsilon}_0$. Then,*

$$\mathbb{E}[\mathrm{M}_2(\boldsymbol{\varepsilon}_1)^2] \leq \mathrm{M}_2(\boldsymbol{\varepsilon}_0)^2 - 2\alpha\xi\,\mathrm{M}_2(\boldsymbol{\varepsilon}_0) + \alpha^2$$

*and the RHS achieves minimum at $\alpha = \xi\,\mathrm{M}_2(\boldsymbol{\varepsilon}_0) \approx \xi\eta$.*

---

[2] As the training proceeds, $\mathbb{E}[\boldsymbol{g}_t]$ will converge to $\approx \boldsymbol{0}$, so Assumption 1 is related to the observation that, for highly expressive models, $\tilde{\boldsymbol{g}}_t = \nabla\ell(z_t; \tilde{\boldsymbol{\theta}}^*)$ can get close to $\boldsymbol{0}$ for every $z_t$ in the training data (Ma et al., 2018). However, Assumption 1 only requires that $\mathrm{M}_2(\boldsymbol{g}_t)$ decreases as fast as $\mathbb{E}[\boldsymbol{g}_t]$, which is empirically verified (§ A.6). Whether $\mathrm{M}_2(\boldsymbol{g}_t)$ actually converges to 0 is not guaranteed (because the training may stop early, or $\mathbb{E}[\boldsymbol{g}_t]$ not get to exactly $\boldsymbol{0}$ due to L2-regularization, etc.) and not used in our theory. On the other hand, $\mathrm{M}_2(\boldsymbol{g}_t)$ is always large compared to $\mathbb{E}[\boldsymbol{g}_t]$, because $\xi$ is a small value.

Lemma 4.3 suggests the initial learning-rate $\alpha_0 = \frac{\xi\eta}{\sqrt{\mathbb{E}[M_2(\boldsymbol{g}_0)^2]}}$. Then, we get $\rho_0 = \frac{1}{2}\xi^2 \frac{M_2(\boldsymbol{g}_0)^2}{\mathbb{E}[M_2(\boldsymbol{g}_0)^2]}$ from Equation 5. By adding the decay factors, we reveal the Amos update rule (Equation 1):

$$\boldsymbol{\delta}_t \leftarrow \alpha_t \boldsymbol{g}_t + \rho_t \boldsymbol{\theta}_t, \text{ where } \alpha_t = d_t \frac{\xi\eta}{\sqrt{\mathbb{E}[M_2(\boldsymbol{g}_t)^2]}} \text{ and } \rho_t = d_t \frac{1}{2}\gamma_t = d_t \frac{1}{2}c_t \xi^2 \frac{M_2(\boldsymbol{g}_t)^2}{\mathbb{E}[M_2(\boldsymbol{g}_t)^2]}. \quad (6)$$

Here, $d_t$ and $c_t$ monotonically decrease to 0 and $d_0 = c_0 = 1$. In particular, $c_t$ decaying to 0 ensures that $\rho_t$ decays to 0 faster than $\alpha_t$, so $\rho_t \boldsymbol{\theta}_t$ can be sufficiently small compared to $\alpha_t \boldsymbol{g}_t$ for large $t$, which justifies the approximation of Equation 4. In § A.2, we will further derive that $c_t = (1 + pb_t)^{-\frac{1}{2}}$ and $d_t = (1 + qb_t)^{-1}$, where $p, q$ are constants, together with the update rule of $b_t$. The specific $p = \frac{1}{4}\sqrt{\xi}$ and $q = \frac{1}{4}\sqrt{\xi\eta}$ are found through experiments and work well in practice.

## 5 EXPERIMENTS

We focus on the Transformer model (Vaswani et al., 2017), and pre-train several variants as below.

**BERT:** A Transformer Encoder model with learned position embeddings (Devlin et al., 2019). We experiment with the base (12-layer 768-hidden) and large (24-layer 1024-hidden) model sizes.

**RoPE:** A Transformer Encoder variant with the Rotary Position Encoding (Su et al., 2021). RoPE is integrated in some recent large-scale language models (Chowdhery et al., 2022). It encodes relative positions but the encoding is not learned. We experiment with the base (12-layer 768-hidden) and large (24-layer 1024-hidden) model sizes.

**Relative Position Embeddings (RPE):** A Transformer Encoder variant with learned relative position embeddings (Shaw et al., 2018). It achieves better performance but the pre-training is more costly on TPU (Tian et al., 2021). We experiment with the base (12-layer 768-hidden) model size.

**T5 Encoder-Decoder (T5):** A Transformer Encoder-Decoder model implemented by Raffel et al. (2020). We experiment with the large (24-layer 1024-hidden) model size.

For encoder only models, we pre-train with the Masked Language Modeling loss (Devlin et al., 2019) on Wikipedia[3] and the Books Corpus (Zhu et al., 2015). Following Liu et al. (2019), we use batch size 1024 for base-sized models, and pre-train 200k or 300k steps. For BERT-large, we use batch size 4096 and pre-train 250k steps. For RoPE-large, due to memory limitations we have to use batch size 1024 and pre-train 1M steps. For T5, the batch size is 4096 and we pre-train with the Span Corruption loss on the C4 corpus (Raffel et al., 2020), for 250k steps. More detailed settings are found in § A.5.

As an additional evaluation, we also applied Amos to the ResNet model (He et al., 2016) on the ImageNet (Deng et al., 2009) dataset. The experiment settings and results are shown in § A.9.

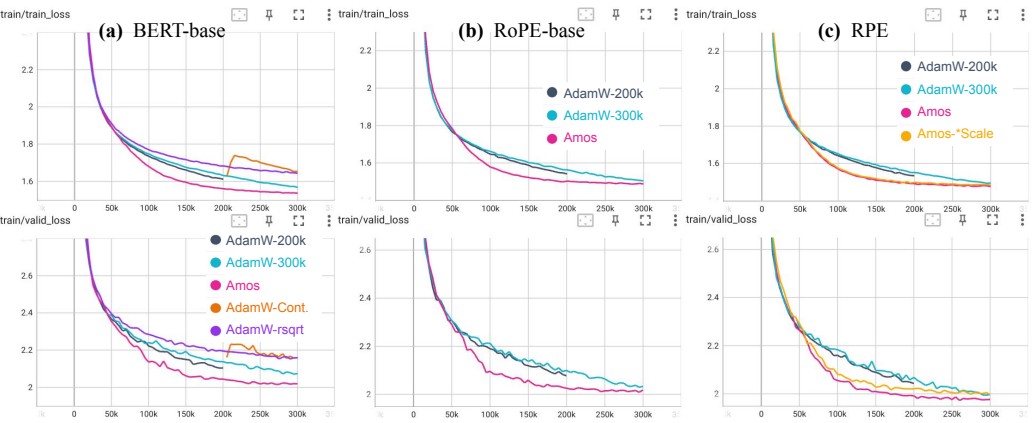

Figure 1: Pre-training 3 models of the base (12-layer 768-hidden) size: (a) BERT, (b) RoPE and (c) RPE. We show training loss on the top and validation loss on the bottom.

---

[3] https://en.wikipedia.org/wiki/Main_Page

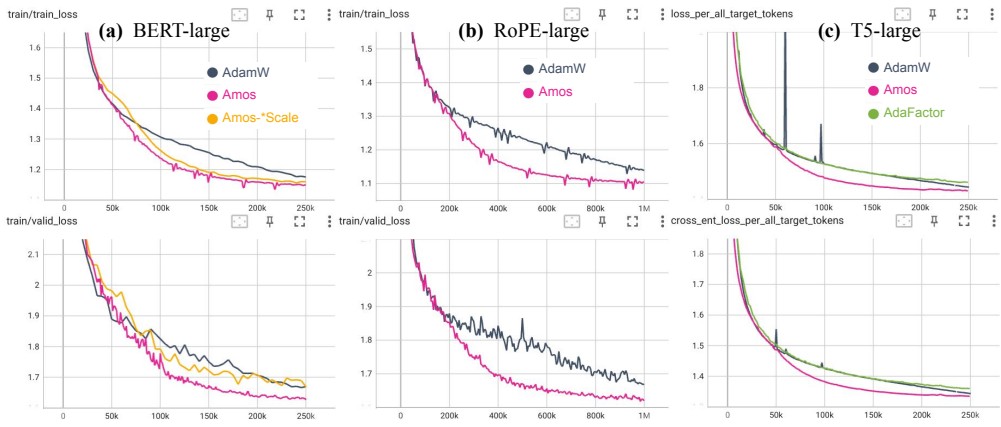

Figure 2: Pre-training 3 models of the large (24-layer 1024-hidden) size: (a) BERT, (b) RoPE and (c) T5. We show training loss on the top and validation loss on the bottom. For T5, the training loss is different from the cross-entropy loss due to an extra regularization term. See § A.5 for details.

## 5.1 LEARNING CURVE OF PRE-TRAINING TRANSFORMER VARIANTS

In Figure 1 and Figure 2, we show training and validation loss of pre-training the Transformer variants. In all experiments, across different model architectures, model sizes, batch sizes, datasets and loss functions, Amos (pink curve) outperforms the state-of-the-art AdamW setting, with the loss always significantly lower beyond 30% of the training procedure[4], and the validation loss achieving the final value of AdamW-300k within $< 70\%$ training steps or time[5]. For BERT-base (Figure 1a), Amos achieves the same within only 145k steps ($< 50\%$), and the Amos checkpoint at 150k outperforms the final checkpoint of AdamW-300k in fine-tuning on MNLI (Williams et al., 2018) as well (§ A.7).

In Figure 1a, we also tried starting from the final checkpoint of AdamW-200k and resetting the learning-rate as if it is linearly decaying to max training step 300k (AdamW-Cont.). The loss spikes higher and does not go further lower than the value at 200k, suggesting that the hyper-parameter of max training steps has to be set a priori, and continuous training is not trivial with AdamW. In addition, we tried a learning-rate schedule (AdamW-rsqrt) that takes the same value at step 10k but adopts a decay in proportion to $t^{-1/2}$ (where $t$ is the step) beyond. Although this setting does not require max training steps, it converges slower than both AdamW-200k and AdamW-300k.

For the RPE model (Figure 1c), we tried setting $\eta$ of the relative position embeddings to a smaller value (Amos-*Scale, see § A.4 for more details), and found significant impact especially on the validation loss. Similar results are observed when we change $\eta$ for a certain type of layers in the BERT-large model (Figure 2a, Amos-*Scale, see § A.4). It suggests that the model-specific

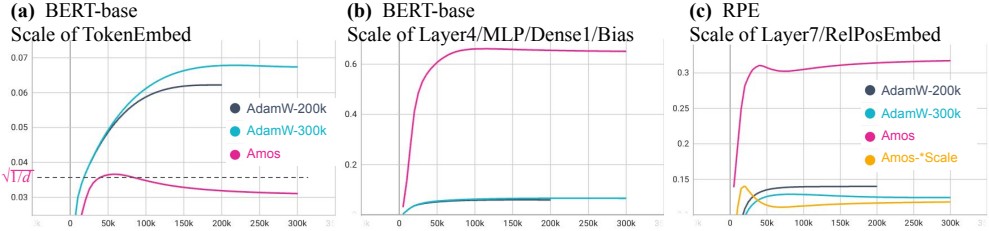

Figure 3: Plots of the quadratic mean of entries of variables over pre-trained steps.

---

[4]We have tried different learning-rates in preliminary experiments and the best was chosen. A learning-rate search for BERT-base is presented in § A.5.

[5]In our JAX (Bradbury et al., 2018) implementation, the running time per training step for all optimizers (AdamW, Amos and AdaFactor) are almost the same.

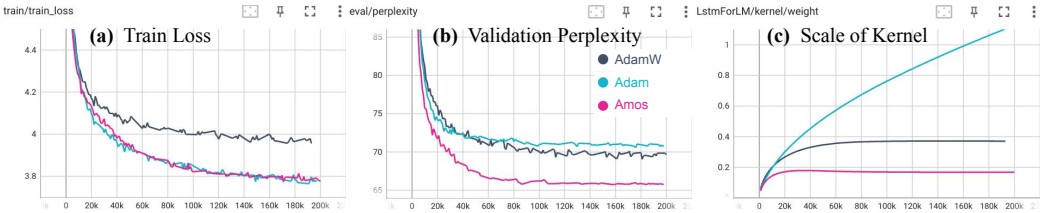

Figure 4: Training a single layer LSTM on the PTB corpus.

information $\tilde{\eta}$ indeed contributes to the performance of Amos, which according to previous work (Kaplan et al., 2020) is unlikely achieved by tuning the learning-rate schedule alone.

## 5.2 Scales of Trained Variables

In Figure 3 we show how the scale of entries of some variables evolve as the training proceeds. With AdamW, both the token embeddings and the bias converge to similar scales (Figure 3ab); while with Amos the token embeddings converge to $\approx \sqrt{1/d}$ (where $d$ is the hidden size) and the bias to $\approx 0.5$, as specified by the hyper-parameter $\tilde{\eta}$. It shows that the algorithm of Amos can lead variables to converge to drastically different scales, which is unlikely with AdamW. In Figure 3c, comparing Amos and Amos-*Scale, the relative position embeddings in a typical layer of the RPE model converge to different scales, which shows that the scale is indeed controlled by the hyper-parameter $\tilde{\eta}$. Recall that Figure 1c shows this has impact on the performance.

In order to further illustrate the relation among the optimizer, validation performance and the scale of variables, we train a single layer LSTM on the Penn Tree Bank (PTB) corpus (Marcus et al., 1993). The model size is 256 for hidden states and 1024 for memory. We set dropout rate 0.55 for hidden states (which is important for training on PTB) and 0.1 for memory. Sequence length and batch size are set to 64. We compare Amos, AdamW, and Adam (without weight decay). For Amos, the global learning-rate is set to 0.01 and $\eta$ for the LSTM kernel is set to $\frac{1}{\sqrt{32}}$ (calculated from input scale $\frac{1}{4}$, input dimension 512 and output scale 1). For AdamW and Adam, the learning-rate is set to 0.0015 (about the same as Amos for the LSTM kernel), and the weight decay is set to 0.01 for AdamW.

The results are shown in Figure 4. Without weight decay, the scale of the LSTM kernel trained by Adam can keep increasing; so Adam is better than AdamW on training loss but worse on validation perplexity (i.e. the model trained by Adam generalizes worse). On the other hand, Amos achieves the same training loss as Adam, while keeping the scale of the kernel as specified. It results in a much better validation perplexity which matches the state-of-the-art[6]. Overall, we conclude that controlling the scale of trained variables can help the generalization performance of deep neural networks, and the model-specific information from $\tilde{\eta}$ enables Amos to do this.

## 6 Conclusion

We have presented the *Amos* optimizer, which uses an adaptive L2 regularizer to control learning-rate decay and guides trained weights towards a specified model-oriented scale. It demonstrates faster convergence than the state-of-the-art in pre-training language models, where the training process is long and decaying schedule is crucial. On the other hand, its ability to control the scale of trained weights also brings better generalization to small models such as a single layer LSTM.

Besides pre-training, we expect Amos to have advantages in fine-tuning as well, especially for multi-modal models that combine heterogeneous components of varied scales and/or pre-trained with different recipes. Hopefully, the model-specific information $\tilde{\eta}$ can help us fine-tune such models that were previously difficult with other optimizers (Liang et al., 2022; Kumar et al., 2022).

---

[6]See Melis et al. (2020) for a setting that achieves the state-of-the-art performance for a single layer LSTM on PTB. It uses RMSProp and dynamically decays the learning-rate by watching the performance on the validation set. To our knowledge, no previous work has been able to achieve the state-of-the-art with a straightforward setting of the optimizer as we do with Amos.

**Ethics Statement**   This work includes pre-training language models, which have the potential risk of inherited bias from the training data. Our empirical contribution is on accelerating the pre-training process and thus does not focus on addressing such risk. For fair comparison, the pre-training data we have used are the same as previous works, and consequently the models we trained to evaluate our approach are similar to those already open-sourced. We refer to Bommasani et al. (2021) for a discussion of the risks of pre-trained language models.

**Reproducibility Statement**   Proof of lemmas in § 4 is given in § A.1. Following the derivation of the Amos update rule, a heuristic derivation of the asymptotic behavior of the Amos decay factors is found in § A.2, and its connection with SGD is discussed in § A.3. Assumption 1 in our derivation is verified by experiments in § A.6. We explain the calculation of $\tilde{\eta}$ for the Transformer models in § A.4. For the pre-training experiments in § 5.1, we describe detailed settings in § A.5, and present a learning-rate search for BERT-base as well. Fine-tuning experiments on MNLI are shown in § A.7. Furthermore, an ablation test for the memory reduction settings of Amos is found in § A.8. Additional experiment settings and results of training the ResNet50 model on ImageNet are found in § A.9. Our code is open-sourced at: `https://anonymous-url`.

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

## A  APPENDIX

### A.1  PROOF OF LEMMAS

*Proof of Lemma 4.1.* We have

$$\mathbb{E}_{t+1}[\ell(z_{t+1}; \tilde{\boldsymbol{\theta}}_{t+1})] = \mathbb{E}_{t+1}[\ell(z_{t+1}; \tilde{\boldsymbol{\theta}}_t - \tilde{\boldsymbol{\alpha}}_t \odot \tilde{\boldsymbol{g}}_t)]$$
$$= \mathbb{E}_{t+1}[\ell(z_{t+1}; \tilde{\boldsymbol{\theta}}_t) - (\tilde{\boldsymbol{\alpha}}_t \odot \tilde{\boldsymbol{g}}_t) \cdot \nabla \ell(z_{t+1}; \tilde{\boldsymbol{\theta}}_t)] + o(\tilde{\boldsymbol{\alpha}}_t),$$

where $\odot$ denotes element-wise multiplication, $o(\tilde{\boldsymbol{\alpha}}_t)/\|\tilde{\boldsymbol{\alpha}}_t\| \to 0$ at $\|\tilde{\boldsymbol{\alpha}}_t\| \to 0$, and arrays are flattened to vectors for the dot-product. Since $z_{t+1}$ and $z_t$ are drawn from the same distribution, we have $\mathbb{E}_{t+1}[\ell(z_{t+1}; \tilde{\boldsymbol{\theta}}_t)] = \mathbb{E}_t[\ell(z_t; \tilde{\boldsymbol{\theta}}_t)]$ and $\mathbb{E}_{t+1}[(\tilde{\boldsymbol{\alpha}}_t \odot \tilde{\boldsymbol{g}}_t) \cdot \nabla \ell(z_{t+1}; \tilde{\boldsymbol{\theta}}_t)] = (\tilde{\boldsymbol{\alpha}}_t \odot \tilde{\boldsymbol{g}}_t) \cdot \mathbb{E}_t[\nabla \ell(z_t; \tilde{\boldsymbol{\theta}}_t)]$. Moreover, because $\tilde{\boldsymbol{\alpha}}_t$ does not depend on $z_t$, we have $\mathbb{E}_t[\tilde{\boldsymbol{\alpha}}_t \odot \tilde{\boldsymbol{g}}_t] = \tilde{\boldsymbol{\alpha}}_t \odot \mathbb{E}_t[\tilde{\boldsymbol{g}}_t]$. Thus,

$$\mathbb{E}_t[\mathbb{E}_{t+1}[\ell(z_{t+1}; \tilde{\boldsymbol{\theta}}_{t+1})]] = \mathbb{E}_t[\ell(z_t; \tilde{\boldsymbol{\theta}}_t)] - (\tilde{\boldsymbol{\alpha}}_t \odot \mathbb{E}_t[\tilde{\boldsymbol{g}}_t]) \cdot \mathbb{E}_t[\nabla \ell(z_t; \tilde{\boldsymbol{\theta}}_t)] + o(\tilde{\boldsymbol{\alpha}}_t).$$

Now $\mathbb{E}_t[\tilde{\boldsymbol{g}}_t] = \mathbb{E}_t[\nabla \ell(z_t; \tilde{\boldsymbol{\theta}}_t)]$ by definition, so $(\tilde{\boldsymbol{\alpha}}_t \odot \mathbb{E}_t[\tilde{\boldsymbol{g}}_t]) \cdot \mathbb{E}_t[\nabla \ell(z_t; \tilde{\boldsymbol{\theta}}_t)] > 0$, and the lemma follows by taking $\tilde{\boldsymbol{\alpha}}_t$ small enough so that $o(\tilde{\boldsymbol{\alpha}}_t)$ can be omitted. □

*Proof of Lemma 4.2.* In the LHS of Equation 5, only $\rho_t$ can depend on $z_t$; while in the RHS, $\mathrm{M}_2(\boldsymbol{g}_t)^2$ depends on $z_t$ but $\alpha_t$ does not. In order to satisfy Equation 5 on every $z_t$, it is necessary that $\rho_t$ has a $\mathrm{M}_2(\boldsymbol{g}_t)^2$ factor: $\rho_t \propto \mathrm{M}_2(\boldsymbol{g}_t)^2$. Moreover, we require that $\mathbb{E}[\rho_t]$ does not depend on $\boldsymbol{g}_t$, so $\rho_t$ should be normalized by $\mathbb{E}[\mathrm{M}_2(\boldsymbol{g}_t)^2]$: $\rho_t \propto \frac{\mathrm{M}_2(\boldsymbol{g}_t)^2}{\mathbb{E}[\mathrm{M}_2(\boldsymbol{g}_t)^2]}$. This, substituted back into Equation 5, implies that $\alpha_t \propto \frac{1}{\sqrt{\mathbb{E}[\mathrm{M}_2(\boldsymbol{g}_t)^2]}}$. $\qquad\square$

*Proof of Lemma 4.3.* Equation 2 implies $\boldsymbol{\varepsilon}_1 = \boldsymbol{\varepsilon}_0 - \alpha_0 \boldsymbol{g}_0$. Taking $\mathbb{E}[\mathrm{M}_2(\bullet)]$ of this equation, we have

$$\mathbb{E}[\mathrm{M}_2(\boldsymbol{\varepsilon}_1)^2] = \mathrm{M}_2(\boldsymbol{\varepsilon}_0)^2 - \frac{2}{k}\alpha_t \mathbb{E}[\boldsymbol{g}_0] \cdot \boldsymbol{\varepsilon}_0 + \alpha_0^2 \mathbb{E}[\mathrm{M}_2(\boldsymbol{g}_0)^2]$$

$$= \mathrm{M}_2(\boldsymbol{\varepsilon}_0)^2 - \frac{2}{k}\alpha \frac{\mathbb{E}[\boldsymbol{g}_0] \cdot \boldsymbol{\varepsilon}_0}{\sqrt{\mathbb{E}[\mathrm{M}_2(\boldsymbol{g}_0)^2]}} + \alpha^2.$$

Since $\boldsymbol{g}_0$ and $\boldsymbol{\varepsilon}_0$ point to the same direction, we have $\frac{1}{k}\mathbb{E}[\boldsymbol{g}_0] \cdot \boldsymbol{\varepsilon}_0 = \mathrm{M}_2(\mathbb{E}[\boldsymbol{g}_0])\,\mathrm{M}_2(\boldsymbol{\varepsilon}_0)$. By Assumption 1 we have $\mathrm{M}_2(\mathbb{E}[\boldsymbol{g}_0])/\sqrt{\mathbb{E}[\mathrm{M}_2(\boldsymbol{g}_0)^2]} \geq \xi$. Hence,

$$\mathbb{E}[\mathrm{M}_2(\boldsymbol{\varepsilon}_1)^2] \leq \mathrm{M}_2(\boldsymbol{\varepsilon}_0)^2 - 2\alpha\xi\,\mathrm{M}_2(\boldsymbol{\varepsilon}_0) + \alpha^2.$$

The RHS above is a quadratic function of $\alpha$, which achieves minimum at $\alpha = \xi\,\mathrm{M}_2(\boldsymbol{\varepsilon}_0)$. Finally, since (usually) $\boldsymbol{\theta}_0$ is initialized close to $\mathbf{0}$, and $\mathrm{M}_2(\boldsymbol{\theta}^*) \approx \eta$, we have $\mathrm{M}_2(\boldsymbol{\varepsilon}_0) \approx \eta$. $\qquad\square$

## A.2 Heuristic Derivation of Decay Factors

Substituting Equation 6 into Equation 5, we get the following equivalent of Equation 5:

$$c_t\,\mathrm{M}_2(\boldsymbol{\varepsilon}_t) = d_t\eta. \tag{7}$$

Without knowing any specific relation among $\boldsymbol{g}_t$, $\boldsymbol{\theta}_t$ and $\boldsymbol{\varepsilon}_t$, we found it difficult to theoretically decide an optimal $c_t$. Given that $c_t$ decreases to $0$, we set $c_t$ to decrease according to $\mathrm{M}_2(\boldsymbol{\varepsilon}_t)$ in Amos, i.e. $c_t \sim r\,\mathrm{M}_2(\boldsymbol{\varepsilon}_t)$, where $r$ is a constant and $\sim$ denotes asymptotically equal at $t \to \infty$. Thus, by Equation 7 we have $d_t \sim \frac{r}{\eta}\,\mathrm{M}_2(\boldsymbol{\varepsilon}_t)^2$. We will analyze the evolution of $\mathrm{M}_2(\boldsymbol{\varepsilon}_t)^2$ to derive $c_t$ and $d_t$.

Taking $\mathbb{E}[\bullet]$ of Equation 4, and applying Equation 6 and Equation 7, we get

$$\mathbb{E}[\mathrm{M}_2(\boldsymbol{\varepsilon}_{t+1})^2] \approx \mathrm{M}_2(\boldsymbol{\varepsilon}_t)^2 - \frac{2}{k}\left(c_t\xi\,\mathrm{M}_2(\boldsymbol{\varepsilon}_t)\frac{\mathbb{E}[\boldsymbol{g}_t]}{\sqrt{\mathbb{E}[\mathrm{M}_2(\boldsymbol{g}_t)^2]}} + \frac{d_t}{2}\mathbb{E}[\gamma_t]\boldsymbol{\theta}_t\right)\cdot\boldsymbol{\varepsilon}_t + c_t^2\xi^2\,\mathrm{M}_2(\boldsymbol{\varepsilon}_t)^2. \tag{8}$$

As in the derivation of the initial learning-rate, we make an optimistic estimation that $\mathbb{E}[\boldsymbol{g}_t]$ and $\boldsymbol{\varepsilon}_t$ have the same direction. Then, applying Assumption 1 we have

$$\frac{1}{k}\frac{\mathbb{E}[\boldsymbol{g}_t]}{\sqrt{\mathbb{E}[\mathrm{M}_2(\boldsymbol{g}_t)^2]}}\cdot\boldsymbol{\varepsilon}_t = \frac{\mathrm{M}_2(\mathbb{E}[\boldsymbol{g}_t])}{\sqrt{\mathbb{E}[\mathrm{M}_2(\boldsymbol{g}_t)^2]}}\,\mathrm{M}_2(\boldsymbol{\varepsilon}_t) \geq \xi\,\mathrm{M}_2(\boldsymbol{\varepsilon}_t),$$

and Equation 8 implies

$$\mathbb{E}[\mathrm{M}_2(\boldsymbol{\varepsilon}_{t+1})^2] \leq \mathrm{M}_2(\boldsymbol{\varepsilon}_t)^2 - 2c_t\xi^2\,\mathrm{M}_2(\boldsymbol{\varepsilon}_t)^2 - \frac{d_t}{k}\mathbb{E}[\gamma_t]\boldsymbol{\theta}_t \cdot \boldsymbol{\varepsilon}_t + c_t^2\xi^2\,\mathrm{M}_2(\boldsymbol{\varepsilon}_t)^2$$

$$\leq \mathrm{M}_2(\boldsymbol{\varepsilon}_t)^2 - c_t\xi^2\,\mathrm{M}_2(\boldsymbol{\varepsilon}_t)^2 - \frac{d_t}{k}\mathbb{E}[\gamma_t]\boldsymbol{\theta}_t \cdot \boldsymbol{\varepsilon}_t \tag{9}$$

where in the last equation we have used the fact that $c_t \leq 1$. Now, in order to estimate $\boldsymbol{\theta}_t \cdot \boldsymbol{\varepsilon}_t$, we assume that $\boldsymbol{\theta}_t$ will be evenly distributed on the hypersphere of radius $\mathrm{M}_2(\boldsymbol{\varepsilon}_t)$ around $\boldsymbol{\theta}^*$ as the training proceeds. Then, if $k \geq 3$, for most $\boldsymbol{\theta}_t$ from the distribution we will have $\boldsymbol{\theta}^* \cdot \boldsymbol{\varepsilon}_t \approx 0$. In this case, we have $\frac{1}{k}\boldsymbol{\theta}_t \cdot \boldsymbol{\varepsilon}_t = \frac{1}{k}(\boldsymbol{\varepsilon}_t + \boldsymbol{\theta}^*) \cdot \boldsymbol{\varepsilon}_t \approx \mathrm{M}_2(\boldsymbol{\varepsilon}_t)^2$, and "on average" it is safe to assume[7] that $\frac{1}{k}\boldsymbol{\theta}_t \cdot \boldsymbol{\varepsilon}_t \geq q\,\mathrm{M}_2(\boldsymbol{\varepsilon}_t)^2$ for some constant $q > 0$. Then, Equation 9 becomes

$$\mathbb{E}[\mathrm{M}_2(\boldsymbol{\varepsilon}_{t+1})^2] \leq \mathrm{M}_2(\boldsymbol{\varepsilon}_t)^2 - \mathbb{E}[\gamma_t](1 + d_t q)\,\mathrm{M}_2(\boldsymbol{\varepsilon}_t)^2 \tag{10}$$

---

[7]It not useful in this work to provide a rigorous definition of "on average". We only point out its deep connection with Stein's example (Stein, 1956) that if $k \geq 3$, an estimator with L2 regularization can be better than the maximum likelihood estimator without L2.

where we have used the fact that $\mathbb{E}[\gamma_t] = c_t\xi^2$. In light of Equation 10, we consider the following asymptotic difference equation:

$$e_{t+1} \sim e_t - \gamma_t(1 + d_tq)e_t \tag{11}$$

where $e_t$ is intended to follow the asymptotic behavior of $\mathrm{M}_2(\boldsymbol{\varepsilon}_t)^2$. Since we have $d_t \sim \frac{r}{\eta}\mathrm{M}_2(\boldsymbol{\varepsilon}_t)^2$, it is natural to assume $d_t \sim \frac{r}{\eta}e_t$. Then, we transform Equation 11 as the following:

$$\frac{1}{e_{t+1}} \sim \frac{1}{e_t} \cdot \frac{1}{1 - \gamma_t(1 + d_tq)} \sim \frac{1}{e_t}\big(1 + \gamma_t(1 + d_tq)\big) \sim \frac{1}{e_t} + \gamma_t\big(\frac{1}{e_t} + \frac{qr}{\eta}\big),$$

in which we have used the approximation $1/(1-x) \sim 1 + x$ applied to $x = \gamma_t(1 + d_tq)$. Thus, the update rule of $b_t$ in Algorithm 1 can be revealed by setting $b_t = \frac{\eta}{qr}\frac{1}{e_t}$:

$$b_{t+1} = b_t + \gamma_t(b_t + 1).$$

And $d_t \sim \frac{r}{\eta}e_t$ implies $d_t \sim \frac{1}{qb_t}$, so we set $d_t = \frac{1}{1 + qb_t}$ to satisfy both the asymptotic behavior and $d_0 = 1$.

Similarly, since $c_t \sim r\,\mathrm{M}_2(\boldsymbol{\varepsilon}_t)$ we have $c_t \sim \frac{1}{\sqrt{pb_t}}$ where $p = \frac{q}{r\eta}$. So we set $c_t = \frac{1}{\sqrt{1 + pb_t}}$ to satisfy the asymptotic behavior and $c_0 = 1$.

## A.3 CONNECTION TO SGD

The derivation of decay factors in Amos (§ A.2) is largely inspired by SGD (Murata, 1998). In this section, we recall the theory of learning-rate schedule of SGD and discuss its relation with Amos.

The update rule of SGD is simply $\delta_t \leftarrow \alpha_t g_t$, where $\alpha_t$ is a scalar learning-rate. It is recommended to set the learning-rate schedule to $\alpha_t = \frac{\alpha}{1+\alpha\lambda t}$, where $\alpha$ is the initial learning-rate and $\lambda$ is the smallest eigen-value of the Hessian (Bottou, 2012). This is based on the following discussion.

**Lemma A.1.** *Assume $\tilde{\boldsymbol{\theta}}_t$ is in a neighborhood of a local minimum $\tilde{\boldsymbol{\theta}}^*$, such that the gradient $\mathbb{E}[\tilde{\boldsymbol{g}}_t]$ is approximated by $\boldsymbol{H}\tilde{\boldsymbol{\varepsilon}}_t$ via Taylor expansion. Here, $\boldsymbol{H} = \mathbb{E}[\nabla^2\ell(z_t; \tilde{\boldsymbol{\theta}}^*)]$ is the Hessian at $\tilde{\boldsymbol{\theta}}^*$. Let $0 < \lambda$ be the smallest eigen-value of $\boldsymbol{H}$. Then,*

$$\mathbb{E}[\mathrm{M}_2(\tilde{\boldsymbol{\varepsilon}}_{t+1})^2] \leq \mathrm{M}_2(\tilde{\boldsymbol{\varepsilon}}_t)^2 - 2\lambda\alpha_t\,\mathrm{M}_2(\tilde{\boldsymbol{\varepsilon}}_t)^2 + \alpha_t^2\mathbb{E}[\mathrm{M}_2(\tilde{\boldsymbol{g}}_t)^2] \tag{12}$$

*and the minimum of RHS of Equation 12 is achieved by*

$$\alpha_t = \frac{\lambda\,\mathrm{M}_2(\tilde{\boldsymbol{\varepsilon}}_t)^2}{\mathbb{E}[\mathrm{M}_2(\tilde{\boldsymbol{g}}_t)^2]} \quad and \quad \mathbb{E}[\mathrm{M}_2(\tilde{\boldsymbol{\varepsilon}}_{t+1})^2] \leq \mathrm{M}_2(\tilde{\boldsymbol{\varepsilon}}_t)^2 - \frac{\lambda^2\,\mathrm{M}_2(\tilde{\boldsymbol{\varepsilon}}_t)^4}{\mathbb{E}[\mathrm{M}_2(\tilde{\boldsymbol{g}}_t)^2]}. \tag{13}$$

*Proof.* Since $\tilde{\boldsymbol{\theta}}^*$ is a local minimum, we have $\mathbb{E}[\nabla\ell(z_t; \tilde{\boldsymbol{\theta}}^*)] = \mathbf{0}$ and $\mathbb{E}[\tilde{\boldsymbol{g}}_t] \approx \boldsymbol{H}\tilde{\boldsymbol{\varepsilon}}_t$, where $\boldsymbol{H}$ is positive definite. Given $\lambda$ the smallest eigen-value of $\boldsymbol{H}$, we have $\mathbb{E}[\tilde{\boldsymbol{g}}_t] \cdot \tilde{\boldsymbol{\varepsilon}}_t \geq \lambda\|\tilde{\boldsymbol{\varepsilon}}_t\|^2$. Applying this to $\mathbb{E}[\mathrm{M}_2(\bullet)]$ of Equation 2, we get Equation 12. Now the RHS is a quadratic function of $\alpha_t$, and it takes minimum at Equation 13. So the lemma follows. $\square$

Note that both Amos and SGD analyze the evolution of $\mathrm{M}_2(\boldsymbol{\varepsilon}_t)^2$ by estimating $\alpha_t g_t \cdot \boldsymbol{\varepsilon}_t$. For SGD this is achieved by approximating $\mathbb{E}[\tilde{\boldsymbol{g}}_t]$ with the Hessian. For Amos, on the other hand, we have to make Assumption 1 due to the gradient normalization factor $1/\sqrt{\mathbb{E}[\mathrm{M}_2(\boldsymbol{g}_t)^2]}$. In both cases, the learning-rate decay is derived by setting $\alpha_t$ in terms of $\mathrm{M}_2(\boldsymbol{\varepsilon}_t)$ so that $\mathrm{M}_2(\boldsymbol{\varepsilon}_t)^2$ decreases fast, then solve the asymptotic behavior of $\mathrm{M}_2(\boldsymbol{\varepsilon}_t)$.

**Heuristic derivation of $\alpha_t$:** We assume $\lim_{t\to\infty}\mathbb{E}[\mathrm{M}_2(\tilde{\boldsymbol{g}}_t)^2] = \nu > 0$. In light of Equation 13, we consider the following asymptotic difference equation:

$$e_{t+1} \sim e_t - \frac{\lambda^2}{\nu}e_t^2 \tag{14}$$

where $e_t$ is intended to follow the asymptotic behavior of $M_2(\tilde{\varepsilon}_t)^2$. We transform Equation 14 as:

$$\frac{1}{e_{t+1}} \sim \frac{1}{e_t} \cdot \frac{1}{1 - \frac{\lambda^2}{\nu} e_t} \sim \frac{1}{e_t}(1 + \frac{\lambda^2}{\nu} e_t) = \frac{1}{e_t} + \frac{\lambda^2}{\nu}$$

so we have $\dfrac{1}{e_t} \sim \dfrac{\lambda^2}{\nu}t$. Now, since $\alpha_t = \dfrac{\lambda \, M_2(\tilde{\varepsilon}_t)^2}{\mathbb{E}[M_2(\tilde{g}_t)^2]}$ we have $\alpha_t \sim \dfrac{\lambda}{\nu} e_t \sim \dfrac{1}{\lambda t}$. So $\alpha_t = \dfrac{\alpha}{1 + \alpha \lambda t}$ satisfies both the asymptotic behavior and $\alpha_0 = \alpha$.

In the above derivation, the assumption $\lim_{t \to \infty} \mathbb{E}[M_2(\tilde{g}_t)^2] = \nu > 0$ states that $\mathbb{E}[M_2(\tilde{g}_t)^2]$ will converge to some non-zero value and will not further decrease. This is often described intuitively as "the stochastic noise of sampled gradients does not vanish", a characteristic feature in the theory of SGD. It is in drastic contrast with Assumption 1: We assume that $\mathbb{E}[M_2(g_t)^2]$ decreases along with $M_2(\mathbb{E}[g_t])$ in Amos. Ma et al. (2018) pointed out that the vanishing of $\mathbb{E}[M_2(g_t)^2]$ might lead to faster convergence; but to our knowledge, Amos is the first work to use the vanishing of $\mathbb{E}[M_2(g_t)^2]$ to actually develop an optimizer that empirically converges faster.

For SGD, the hyper-parameter $\lambda$ is generally unknown; but if we adopt an L2 regularizer of strength $\lambda'$, it is guaranteed that $\lambda \geq \lambda'$, so one can safely set the learning-rate to $\frac{\alpha}{1 + \alpha \lambda' t}$ (Bottou, 2012). In Amos, the strength of L2 regularization $\tilde{\gamma}_t$ takes a similar role in controlling the speed of learning-rate decay. We expect this work to inspire more theoretical investigation into this principle.

## A.4 THE CALCULATION OF $\tilde{\eta}$

As explained in §2, for a linear transformation $y = xW + u$ ($W, u \subseteq \tilde{\theta}$, $W \in \mathbb{R}^{m \times n}$, $x \in \mathbb{R}^m$), we set $\eta(W) = \sigma_y/(\sigma_x \sqrt{m})$ and $\eta(u) = \sigma_y/2$, where $\sigma_x$ is the standard deviation of entries of $x$ and $\sigma_y$ the standard deviation of entries of $y$. The values of $\sigma_x$ and $\sigma_y$ are constrained by connected layers, and non-linear layers usually expect entries of input/output tensors from some approximate range. In Table 1, we show 3 types of non-linear layers that occur in Transformer, and specify their input/output range (i.e. expected standard deviation) used for calculating $\tilde{\eta}$.

For activations, e.g. GELU (Hendrycks & Gimpel, 2016) in the Multi-Layer Perceptron (MLP) block, the input range is set to 1 because the non-linearity of the activation function mostly lies within that range; and the output range is set to $\sqrt{1/2}$ because the activation function, as similar to ReLU (Nair & Hinton, 2010), will map negative values (which account for $1/2$ of the input dimension) to close to 0 and approximately retain positive values.

For Softmax of $n$ classes, the input range is set to 1 because the derivative of $\exp(x)$ is close to 1 within the $|x| \leq 1$ range (so Softmax is most sensitive to values within this range); and the output range is set to $\sqrt{1/n}$ because the output is an $n$-dimension vector of L2 norm $\leq 1$ (so the quadratic mean of entries $\leq \sqrt{1/n}$).

For LayerNormalization (Ba et al., 2016), the input range is arbitrary because the input will be normalized. The output range is expected to be 1.

We will discuss the calculation of $\tilde{\eta}$ for specific models in the next sub-sections.

### A.4.1 BERT, RoPE AND RPE

For BERT, RoPE and RPE, the multi-headed attention layer receives the hidden state $x$, and the linear transformations $xQ$ (i.e. the query) and $xK$ (i.e. the key) are expected to have standard deviation 1 so that the dot-product $\sqrt{1/h}(xQ) \cdot (xK)$ (i.e. attention score) has standard deviation 1 as well

| Type of Non-linear Layer | Input Range | Ourtput Range |
|---|---|---|
| Activation in MLP | 1 | $\sqrt{1/2}$ |
| Softmax of $n$ classes | 1 | $\sqrt{1/n}$ |
| LayerNormalization | N/A | 1 |

Table 1: The input/output range of non-linear layers we specify in this work for calculating $\tilde{\eta}$.

| Type of Variable | $\eta$ | Remark |
|---|---|---|
| Bias in all Linears | 0.5 | |
| LayerNormalization Scale | 1 | |
| Input Embeddings | $\sqrt{1/d}$ | $d$ is the size of hidden states |
| MLP/Dense2/Kernel | $\sqrt{2/m}$ | $m$ is the size of intermediate activation in the MLP |
| Other Linear Kernels | $\sqrt{1/d}$ | $d$ is the size of hidden states |
| Relative Position Embeddings | 0.5 | |

Table 2: The $\eta$ calculated for variables in BERT, RoPE and RPE. MLP/Dense2/Kernel is the linear kernel for the output layer of the MLP block. Other linear kernels include e.g. query, key and value kernels in the multi-headed attention layer.

(and this is why there is the scaling factor $\sqrt{1/h}$, where $h$ is the size per-head), which is expected by the Softmax for calculating the attention probability. Therefore, the output ranges of $\boldsymbol{Q}$ and $\boldsymbol{K}$ are 1. For RoPE, the dot-product is replaced by a bi-linear form which encodes relative positions, but this does not change the scale because the bi-linear form is orthogonal.

For other linear transformations in the model, the outputs are either fed into the activation function of an MLP (which requires input range 1), or serve as a summand in a Residual Connection where the residual part comes from a LayerNormalization (which has range 1). So *all the linear transformations have output range 1* in these model architectures.

Thus, we set the $\eta$ of bias in all linear transformations to 0.5, and the $\eta$ for kernels is categorized by the input range and dimension, as we show in Table 2.

The input embeddings (i.e. token embeddings, position embeddings and segment-type embeddings) are inputs to LayerNormalization so their scales are not constrained there; but the token embeddings are also used as the linear kernel for producing the logits of token generation, which expects input range 1 (because it comes from LayerNormalization) and input dimension $d$ (where $d$ is the hidden size), so $\eta$ is set to $\sqrt{1/d}$.

For the linear kernel of the MLP output layer (MLP/Dense2/Kernel), the input range is $\sqrt{1/2}$ because it comes from a non-linear activation, and input dimension $m$ is the size of intermediate activation in the MLP, so $\eta$ is $\sqrt{2/m}$.

For all other linear kernels, the input range is 1 because it comes from LayerNormalization, and input dimension is the hidden size $d$. So $\eta$ is $\sqrt{1/d}$.

The relative position embeddings in the RPE model is used as input to the key and value transformations at each layer, similar to the hidden state. We set $\eta$ to 0.5 so its scale is close to the hidden state (which has scale 1) but will not dominate it.

**Experiments with Amos-*Scale** In § 5.1, we have experimented with pre-training RPE and BERT-large with different $\tilde{\boldsymbol{\eta}}$ (Amos-*Scale). For RPE (Figure 1c), we tried setting $\eta$ of the relative

| Type of Variables | $\eta$ | Remark |
|---|---|---|
| LayerNormalization Scale | 1 | |
| Query Kernel | $\sqrt{1/(hd)}$ | $h$ is the size per-head and $d$ is the size of hidden states |
| Input Embeddings | 1 | |
| MLP/wo/Kernel | $\sqrt{2/m}$ | $m$ is the size of intermediate activation in the MLP |
| Other Linear Kernels | $\sqrt{1/d}$ | $d$ is the size of hidden states |
| Relative Attention Bias | 0.5 | |

Table 3: The $\eta$ calculated for variables in T5. MLP/wo/Kernel is the linear kernel for the output layer of the MLP block.

position embeddings to $\sqrt{1/d}$ instead of $0.5$. For BERT-large (Figure 2a), we tried setting $\eta$ of MLP/Dense2/Kernel to $\sqrt{1/d}$ instead of $\sqrt{2/m}$. They both had impact on performance. Especially for BERT-large, $\sqrt{1/d}$ and $\sqrt{2/m}$ only differ by a $\sqrt{2}$ factor (because $m = 4d$), still the performance gap is significant. It illustrates the importance of setting $\tilde{\boldsymbol{\eta}}$ appropriately.

### A.4.2 T5

For the T5 model, $\eta$ is set as in Table 3. It is different from Table 2, due to several differences between the T5 architecture and BERT, as discussed below.

1. Linear transformations do not have bias terms in T5.
2. Attention score is calculated by $(\boldsymbol{x}\boldsymbol{Q}) \cdot (\boldsymbol{x}\boldsymbol{K})$ in T5, without the scaling factor. Instead, the query kernel $\boldsymbol{Q}$ is initialized to a smaller scale $\sqrt{1/(hd)}$, with an extra $\sqrt{1/h}$ factor compared to $\boldsymbol{K}$. Thus, we accordingly set $\eta$ of the query kernel to $\sqrt{1/(hd)}$.
3. The token embeddings are no longer re-used for producing logits of token generation. So we set $\eta$ to $1$, which is the same as the scale for initialization.
4. The MLP activation function (i.e. gated-GELU) used in T5 is different from BERT. Still, $\eta$ for the linear kernel of the MLP output (MLP/wo/Kernel) is set to the same.
5. We set $\eta$ of the relative attention bias to $0.5$ so its scale is close to the attention score (which has scale 1) but will not dominate it.

### A.5 Detailed Experiment Settings and Learning-rate Search

In this section, we discuss detailed settings of the pre-training experiments in § 5.1. The hyper-parameters and required computation resources are shown in Table 4. For pre-training BERT with AdamW, we follow the settings of Liu et al. (2019). For RPE, pre-training on TPU is slow, so we use a different configuration with more TPU cores to train the base-sized model. For T5, we found that using $\beta = 0.98$ for Amos and AdamW causes training instability, so we decrease the value to $\beta = 0.95$. The settings of AdaFactor follow Raffel et al. (2020) and Shazeer & Stern (2018).

For encoder-only models (i.e. BERT, RoPE and RPE) trained on the Wikipedia+Books corpus, we use the Penn TreeBank corpus (Marcus et al., 1993) as the validation set. The training precision is float32. Number of warm-up steps is set to 10k for AdamW and 20k for Amos.

For T5, the training loss is cross-entropy with an extra regularization term, $(\log Z)^2$ (where $Z$ is the normalization factor in Softmax), which makes the logits close to mean 0 and self-normalized. In

|  | Batch Size | Optimizer | $\beta$ | Learning-rate | #Steps | Resource |
|---|---|---|---|---|---|---|
| BERT-base | 1024 | AdamW | 0.98 | 2e-4 | 200k/300k | TPUv4 2x2x4 About 2 days |
|  |  | Amos | 0.98 | 0.01 | 300k |  |
| RoPE-base | 1024 | AdamW | 0.98 | 2e-4 | 200k/300k |  |
|  |  | Amos | 0.98 | 0.01 | 300k |  |
| RPE | 1024 | AdamW | 0.98 | 2e-4 | 200k/300k | TPUv3 8x8 About 4 days |
|  |  | Amos | 0.98 | 0.01 | 300k |  |
| BERT-large | 4096 | AdamW | 0.98 | 2e-4 | 250k | TPUv4 4x4x4 About 4 days |
|  |  | Amos | 0.98 | 0.01 | 250k |  |
| RoPE-large | 1024 | AdamW | 0.99 | 1e-4 | 1M |  |
|  |  | Amos | 0.99 | 5e-3 | 1M |  |
| T5-large | 4096 | AdamW | 0.95 | 1e-3 | 250k |  |
|  |  | Amos | 0.95 | 0.01 | 250k |  |
|  |  | AdaFactor | 0.8 | 0.01 | 250k |  |

Table 4: Hyper-parameter settings and required computational resources. The hyper-parameter $\beta$ in Amos is corresponding to $\beta_2$ in AdamW and the (second moment) decay rate in AdaFactor.

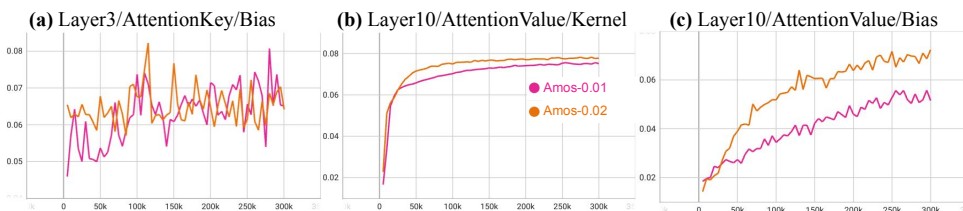

Figure 5: Validation loss for pre-training BERT-base. We compare different learning-rates for (a) AdamW with batch size 1024, (b) Amos with batch size 1024 and (c) Amos with batch size 256.

Figure 6: Plot of the ratio $\frac{M_2(\mathbb{E}[\boldsymbol{g}_t])}{\sqrt{\mathbb{E}[M_2(\boldsymbol{g}_t)^2]}}$ for variables in the BERT-base model, over pre-training steps.

Figure 2c, we plot cross-entropy for validation loss instead of the loss used for training. The training precision of T5 is bfloat16. Possibly because linear transformations in T5 do not have bias terms, we found the model easier to train than BERT, and Amos can be applied without warm-up of learning-rate. The number of warm-up steps is set to 10k for both AdamW and AdaFactor. Learning-rate decay is in proportion to $t^{-1/2}$ (where $t$ is the step) for AdaFactor and linear for AdamW.

For pre-training BERT-base, we present a learning-rate search in Figure 5. For AdamW (Figure 5a), a smaller learning-rate significantly slows down the convergence, while a larger one results in a bumpy validation loss but almost the same performance. On the other hand, both smaller or larger learning-rate can degrade performance for Amos (Figure 5bc). Comparing Figure 5b and Figure 5c, we also verify a theoretical prediction about the global learning-rate of Amos in §4, i.e. the best learning-rate for Amos is in proportion to the square-root of the batch size: Training with $4\times$ the batch size matches $2\times$ the learning-rate.

## A.6 VERIFICATION OF ASSUMPTION 1

In Assumption 1, we have assumed that $\frac{M_2(\mathbb{E}[\boldsymbol{g}_t])}{\sqrt{\mathbb{E}[M_2(\boldsymbol{g}_t)^2]}} \geq \xi > 0$ for all $t$ and across all variables. $\mathbb{E}[\boldsymbol{g}_t]$ and $\mathbb{E}[M_2(\boldsymbol{g}_t)^2]$ can be estimated by taking the running average of $\boldsymbol{g}_t$ and $M_2(\boldsymbol{g}_t)^2$, respectively; so in Figure 6 we track the pre-training of the BERT-base model, calculate the running averages with exponential decay rate $0.98$, and show some typical plots of the ratio. We note two characteristics of the plots: (1) the ratios are *increasing* as the training proceeds, which suggests that taking a global constant $\xi$ to satisfy Assumption 1 is indeed possible; (2) starting points on the left of these plots are similar across different learning rates, which suggests that it is detectable in the early stage of training whether a learning-rate is too small or too large. In fact, in all plots for all variables we can see that the ratio $\frac{M_2(\mathbb{E}[\boldsymbol{g}_t])}{\sqrt{\mathbb{E}[M_2(\boldsymbol{g}_t)^2]}} \geq 0.01$; the appropriate global learning-rate can be read from these plots.

## A.7 FINE-TUNING RESULTS

In Table 5, we show fine-tuning results on the MNLI (Williams et al., 2018) dataset. We compare checkpoints pre-trained for 150k and 300k steps with Amos, and the final checkpoints of AdamW-200k and AdamW-300k. We fine-tune all checkpoints using the Adam optimizer with learning-rate 5e-6, batch size 16, and evaluate by the best accuracy on the MNLI dev set among every 1k of 200k training steps. We run each experiment 3 times and report the mean and standard deviation. The checkpoint pre-trained for 150k by Amos already outperforms the final checkpoint of AdamW-300k.

|              | MNLI-matched   | MNLI-mismatched |
|--------------|----------------|-----------------|
| Amos@150k    | $84.15 \pm .40$ | $84.17 \pm .37$ |
| Amos@300k    | $84.72 \pm .15$ | $84.44 \pm .26$ |
| AdamW-200k   | $83.19 \pm .37$ | $83.45 \pm .41$ |
| AdamW-300k   | $83.84 \pm .14$ | $83.88 \pm .17$ |

Table 5: Fine-tuned accuracy on MNLI dev set. We show the mean and standard deviation of 3 runs.

Thus, the faster convergence by Amos in pre-training indeed transfers to better performance in fine-tuning; we can save $50\%$ of the pre-training cost by using Amos instead of AdamW.

### A.8 ABLATION OF MEMORY REDUCTION

In this section, we experiment with different settings of the memory reduction. We compare the current setting of reducing the input dimension for linear transformations (Reduce_1Axis), to no memory reduction at all (No_Reduce), and the setting of reducing both axes for linear transformations (Reduce_Dense). For embedding matrices, no axis is reduced in the No_Reduce setting, and the embed dimension is reduced for both Reduce_1Axis and Reduce_Dense. We have tried reducing both axes for embedding matrices as well, but found the training unstable in this setting. The comparison of memory usage for slot variables is shown below.

$$\text{AdaFactor (No Momentum)} \ll \text{Reduce\_Dense} < \text{Reduce\_1Axis} \ll \text{AdamW} \ll \text{No\_Reduce}.$$

Without memory reduction, Amos (No_Reduce) consumes more memory than AdamW because it has more slot variables ($\tilde{v}_t, \tilde{b}_t, \tilde{m}_t$ vs. $\tilde{v}_t, \tilde{m}_t$). When memory reduction is applied, the memory usage of $\tilde{v}_t, \tilde{b}_t$ becomes negligible compared to the momentum $\tilde{m}_t$, so Amos (Reduce_1Axis and Reduce_Dense) requires $< 51\%$ memory for slot variables than AdamW. The memory reduction method used by Amos is more efficient than the matrix factorization used by AdaFactor, but in the pre-training of T5 (Figure 2c), AdaFactor achieved favorable performance (although slightly worse in the end than AdamW with linear learning-rate decay) without using momentum, reducing the memory usage further. Whether Amos can achieve a similar performance without using momentum is unclear yet.

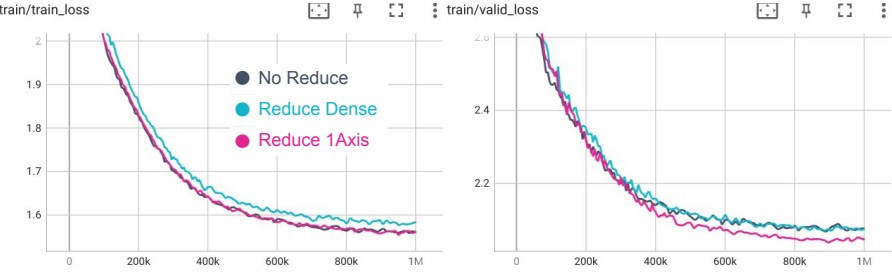

Figure 7: Pre-training BERT-base using Amos with different memory reduction settings.

In Figure 7, we show the training and validation loss of pre-training BERT-base by Amos with different memory reduction settings. Reduce_Dense is slightly worse in training loss compared to No_Reduce, but not so much in validation loss. On the other hand, Reduce_1Axis is almost the same as No_Reduce in training loss, and generalizes even slightly better in validation loss than the other two. So the current Reduce_1Axis setting for Amos is favorable.

### A.9 TRAINING RESNET50 ON IMAGENET

In this section, we apply Amos to the training of ResNet50 (He et al., 2016) on the ImageNet dataset (Deng et al., 2009). ResNet50 is a deep Convolutional Neural Network of 50 layers, with Batch Normalization (Ioffe & Szegedy, 2015) and Residual Connection. ImageNet is a 1000-class image

| Type of Non-linear Layer | Input Range | Ourtput Range |
|---|---|---|
| ReLU Activation | 1 | $\sqrt{1/2}$ |
| BatchNormalization | N/A | 1 |
| Max-pooling on patch size $n$ | 1 | $1/\sqrt{2\ln n}$ |

Table 6: The input/output range of non-linear layers we use to calculate $\tilde{\eta}$ for ResNet.

classification task with 1.28M traning examples. We train with batch size 1024, on an 8-core TPU machine. The settings for Amos is out-of-the-box: the hyper-parameter $\beta$ is set to 0.95, warmup steps 5k, and the global learning rate $\xi$ is set to $\frac{1}{\sqrt{N}} = 0.028$, where $N = 1281167/1024$ is the number of batches in the traning data. We use the open-sourced `init2wint`[8] codebase to run the experiments.

### A.9.1 THE CALCULATION OF $\tilde{\eta}$ FOR RESNET

In order to calculate the hyper-parameter $\tilde{\eta}$ for ResNet, we specify the input/output range of 3 types of non-linear layers in Table 6. This is similar to Transformers, with the only specialty that the output range of a Max-pooling layer is set to $1/\sqrt{2\ln n}$, where $n$ is the patch size. This is because the maximum of $n$ normally distributed random variables[9] has a standard deviation of about $1/\sqrt{2\ln n}$.

The calculated $\eta$ for different types of variables in ResNet is shown in Table 7.

BatchNormalization is treated the same as LayerNormalization in Transformer.

The projection kernel of the first residual block is scaled up by $\sqrt{2\ln n}$ because of its previous max-pooling layer of patch size $n$.

The 2nd and 3rd convolution kernels in each residual block is scaled up by $\sqrt{2}$ because their inputs come from a ReLU activation.

The variables for bias and other linear kernels are treated the same as in Transformer.

**Settings of Amos-\*Scale** We also tried an Amos-\*Scale setting where the $\eta$ for the projection kernel of the first residual block is set to $\sqrt{1/d}$ instead of $\sqrt{(2\ln n)/d}$ (in ResNet50, $n = 3 \times 3 = 9$).

### A.9.2 RESULTS

In Figure 8a, we show the validation error rate of Amos and Amos-\*Scale, where the error rate for Amos (0.261 lowest) is slightly better than the Amos-\*Scale setting (0.263 lowest). Furthermore, it is known that a strong L2 regularization is beneficial for many popular image classification tasks (Loshchilov & Hutter, 2019), but Amos does not have a hyper-parameter to adjust the strength of L2

| Type of Variable | $\eta$ | Remark |
|---|---|---|
| Bias | 0.5 | |
| BatchNormalization scale | 1 | |
| Projection kernel of the first residual block | $\sqrt{(2\ln n)/d}$ | $n$ is the patch size of the previous max-pooling; $d$ is the input size |
| The 2nd and 3rd convolution kernels in each residual block | $\sqrt{2/d}$ | $d$ is the input size |
| Other linear kernels | $\sqrt{1/d}$ | $d$ is the input size |

Table 7: The $\eta$ calculated for variables in ResNet. Other linear kernels include convolution kernels and the final linear classification kernel.

---

[8] https://github.com/google/init2winit
[9] https://en.wikipedia.org/wiki/Fisher%E2%80%93Tippett%E2%80%93Gnedenko_theorem#Gumbel_distribution

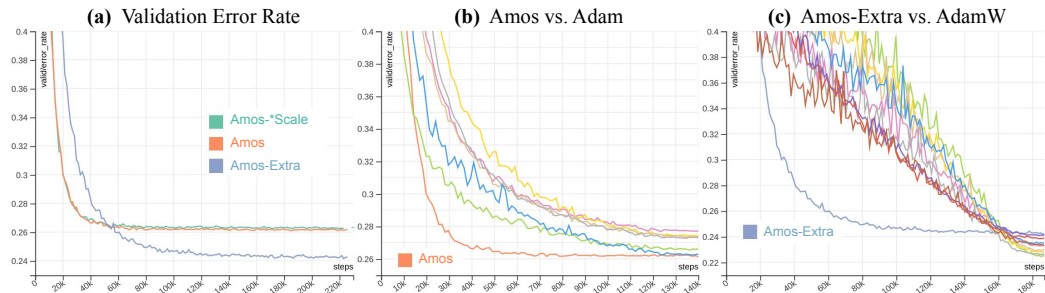

Figure 8: Training ResNet50 on ImageNet. We plot error rate of the validation set.

regularization; so we tried an ad hoc setting Amos-Extra, where the Amos update rule (Equation 1) is replaced by $\boldsymbol{\delta}_t \leftarrow d_t \left( \frac{\xi\eta}{\sqrt{\hat{v}_t}} \boldsymbol{g}_t + (\frac{1}{2}\gamma_t + 0.001)\boldsymbol{\theta}_t \right)$ with everything else kept the same (we also tried other constants, but 0.001 was the best). As shown in Figure 8a, Amos-Extra (0.242 lowest error rate) significantly improves the performance on ImageNet.

In Figure 8b, we compare the out-of-the-box Amos with Adam (no weight decay). The learning-rate schedule of Adam is set to cosine decay with 5% warmup, and the number of training steps is set to 140k. The base learning-rate is tuned by a random search of log scale between 1e-5 and 1e-2, with 25 runs. Other hyper-parameters are set to the default (i.e. $\beta_1 = 0.9$ and $\beta_2 = 0.999$). Amos outperforms all the 25 runs; the best 6 of the 25 are shown in Figure 8b. As alternative settings for Amos, we have also tried $\beta = 0.98$, 0.999, or $\xi = 0.02$, or even changed the decay factors to $c_t = \left(1 + \frac{1}{16}\sqrt{\xi}b_t\right)^{-\frac{1}{2}}$ and $d_t = \left(1 + \frac{1}{16}\sqrt{\xi\eta}b_t\right)^{-1}$. All the other settings converge to almost the same validation error rate, sometimes with slightly slower convergence.

In Figure 8c, we compare Amos-Extra with the state-of-the-art settings of AdamW. The learning-rate schedule of AdamW is set to cosine decay with 5% warmup, and the number of training steps is set to 187k. The base learning-rate, weight decay strength, and label smoothing rate (defaults to 0.1 for other experiments) are tuned by random search, of log scale between 1e-4 and 1e-2, log scale between 1e-2 and 1.0, and linear scale between 0.0 and 0.2, respectively, with 25 runs. Other hyper-parameters are set to the default (i.e. $\beta_1 = 0.9$ and $\beta_2 = 0.999$). Among the 25 runs, 9 of them outperform Amos-Extra, which are shown in Figure 8c. The best performing settings of AdamW gain their advantage close to the end of training, which is probably due to the interaction between weight decay and cosine learning-rate schedule. On the other hand, Amos-Extra demonstrates faster and more stable convergence.

To conclude, when applied to ResNet50 on ImageNet, Amos can outperform Adam out-of-the-box, and become comparable to the state-of-the-art AdamW settings by adding a small constant weight decay term. However, the extra weight decay term is ad hoc, cannot be covered by our current theory (because we have assumed that the L2 regularization is weak enough and decays to 0, not to bias the loss function but only constrain the scale of trained variables), and probably is not the optimal way to strengthen L2 regularization. It leaves the problem of searching for a more general working theory that enables stronger L2 to future work.

