# OpenReview forum: "Amos: An Adam-style Optimizer with Adaptive Weight Decay towards Model-Oriented Scale"
_ICLR.cc/2023/Conference — Submitted to ICLR 2023_

### Official Review · Reviewer_Fga8 · 2022-10-24

**Confidence:** 4
**Correctness:** 2
**Technical Novelty And Significance:** 3
**Empirical Novelty And Significance:** 2
**Recommendation:** 5

**Clarity, Quality, Novelty And Reproducibility:**

The wring and presentation is not good, and there are too many heuristic derivations without further details. However, if all the heuristic parts are right, the novelty is enough. Since how to adaptively adjust the LR and weight decay with theoretical guarantee remains open.

**Strength And Weaknesses:**

#### Strength
- The proposed optimizer can adaptively choose the decaying schedules for learning rate the L-2 regularize coefficient during training.
- There are fewer sensitive hyper-parameters compared to the previous optimizers.

#### Weaknesses
-  Since the optimizer is not explicitly designed for the language model, the absent experimental results on other tasks, such as the vision or RL tasks, degenerate the contribution of this optimizer.
- Some typos, e.g., step one in Algorithm 1, in which the $g_t$ should be bold. The bold and normal formats are used at will in the $\alpha_t$ below Eq.(2).
- Actually, $\hat{v}_t$ is not nearly equal to $E[M_2(g_t)^2]$. In general, we have $$(\hat{v}_t - E[M_2(g_t)^2])^2 \leq \beta (\hat{v}_t - E[M_2(g_{t-1})^2])^2 + \frac{\beta^2}{1-\beta}(E[M_2(g_t)^2] - E[M_2(g_{t-1})^2])^2$$ (see [1], Lemma 2), which is large until convergence. Hence it is strange to replace  $\hat{v}_t$ with $E[M_2(g_t)^2]$.
- The approximation in Eq.(4) is confusing. Since $\alpha_t g_t$ can be small when we are close to convergence, but $\rho_t \theta_t$ is not. We can not ignore $\rho_t \theta_t$ directly.
- Why $\rho_t $  should be normalized by $E[M_2(g_t)^2]$? The stated reason ``the L2-regularizer should (on average) not depend on the norm of $g_t$'' seems weak and not solid.
- Lack the comparison to some strong baselines, such as LAMB and Adam's variants.


[1] Stochastic compositional gradient descent: algorithms for minimizing compositions of expected-value functions.

**Summary Of The Paper:**

The submission presents a new optimizer, Amos, which can heuristically determine the initial learning rate and adjust its decaying schedules for LR and L2 regularizer during training. The experiments show that Amos converges faster than AdamW on language models.

**Summary Of The Review:**

Due to the missing (1) experimental results on general tasks, (2) lack of convergence guarantee, and (3) too many confusing mathematic details in derivations, currently, I hold a negative score on the paper.

---

> ### Author Response · Authors · 2022-11-19
> **We have rewritten the derivation section and added ImageNet experiments; Reply with clarification and welcome discussion**
>
> We appreciate the reviewer comments, and have revised the paper significantly (clearer derivation, ImageNet experiment, etc.). Specific answers are provided below.
>
> * _absent experimental results on other tasks_
>
> We have added experiments with ResNet50 on ImageNet, and Amos works quite well out-of-the-box (Section A.9).
>
> * _The bold and normal formats are used at will_
>
> Scalars and vectors are mixed in our equations due to the memory reduction mechanism, which is somehow the nature of this work; we have added more explanations in the revised paper.
>
> * _Actually,_ $\hat{v}_t$ _is not nearly equal to_ $\mathbb{E}[M_2(g_t)^2]$
>
> Thanks for bringing up this inequality:
>
> $\left(\hat{v}_t - \mathbb{E}[M_2(g_t)^2]\right)^2$
>
> $\quad\leq\beta\big(\hat{v}_{t}$
>
> $\quad\quad - \mathbb{E}[M_{2}(g_{t-1})^2]\big)^2$
>
> $\quad\quad+ \frac{\beta^2}{1-\beta}\left(\mathbb{E}[M_2(g_t)^2] - \mathbb{E}[M_2(g_{t-1})^2]\right)^2$
>
> It is a nice inequality. So this inequality implies that, when $\mathbb{E}[M_2(g_t)^2]$ is changing slowly, i.e. if $\left(\mathbb{E}[M_2(g_t)^2] - \mathbb{E}[M_2(g_{t-1})^2]\right)^2$ is small, then $\hat{v}_t$ will converge to $\mathbb{E}[M_2(g_t)^2]$ within about $1/(1-\beta)$ steps. So if we use $\beta=0.98$, it means $\hat{v}_t$ will converge to $\mathbb{E}[M_2(g_t)^2]$ within about 50 steps, which justifies that $\mathbb{E}[M_2(g_t)^2]$ can be approximated by $\hat{v}_t$.
>
> Indeed, $\hat{v}_t$ is intended as an approximation for $\mathbb{E}[M_2(g_t)^2]$ in the Adam paper; and in the AdaFactor paper they have tested this by experiments (see their Figure 1).
>
> * _Since $\alpha_t g_t$ can be small when we are close to convergence, but $\rho_t \theta_t$ is not_
>
> In Amos, $\rho_t$ is decaying to 0 faster than $\alpha_t$, so $\rho_t \theta_t$ can be smaller than $\alpha_t g_t$. We made this clear in the revision.
>
> * _Why $\rho_t$ should be normalized by $\mathbb{E}[M_2(g_t)^2]$?_
>
> Because on average the L2-regularization should have the same strength across all variables, rather than be affected by the typical gradient magnitude on each variable. It is actually the same intuition that motivates the decoupled weight decay in the AdamW paper. We made this clear in the revision.
>
> * _Lack the comparison to some strong baselines, such as LAMB and Adam's variants._
>
> We are comparing with the state-of-the-art settings of AdamW and AdaFactor. Comparison with LAMB in extremely-large-batch-size settings is left to future work.

---

### Official Review · Reviewer_f9Uo · 2022-10-25

**Confidence:** 3
**Correctness:** 2
**Technical Novelty And Significance:** 2
**Empirical Novelty And Significance:** 3
**Recommendation:** 3

**Clarity, Quality, Novelty And Reproducibility:**

**Quality:** While the method achieves very convincing experimental results, the _scientific_ quality of the paper is subpar in my opinion. The heuristic derivation involves so many approximations, heuristics, and "intuitive arguments" that I found it more confusing and obfuscating than helpful. Some of my specific concerns are listed above (weakness 2). Even more importantly, the method combines various, largely independent, innovations and it is entirely unclear to me, which of these actually contribute to the experimental success (weakness 1). In terms of experiments, the investigation on Transformers is extensive, but I would have expected to see experiments involving, e.g., ResNets on some vision data, for breadth (weakness 3)

**Clarity:** The paper is generally well-written. Some of the steps in the derivation are not well-motivated in my opinion (weakness 2).

**Originality:** The method proposes several innovations compared to Adam that are definitely novel. I think the general direction of using structural knowledge about the model (here: expected scales of the optimal weights) is a very promising yet under-explored.

**Strength And Weaknesses:**

### Strengths

1. Improving optimization algorithms and/or estimating their hyperparameters using structural knowledge of the model is a very promising research direction. To my knowledge, this is one of only a few papers that uses model characteristics to inform the optimization hyperparameters (beyond layer-wise normalization and/or step sizes).

2. The experiments on Transformers are extensive and consistently demonstrate significant improvements over state-of-the-art competitors. Given the importance and immense cost of such training tasks, that is a very significant improvement.

### Weaknesses

1. My main criticism is that the method combines a large number of aspects without proper ablations. If a method gives very good results, but we can't pin down which innovation really drives that improvement, the scientific value is greatly diminished. Comparing Eq. (1) to Adam or RMSProp, I can see at least 6 components/innovations that could be seen as largely independent and I would like to see investigated in isolation
    - The factored learning rate involving the "model-oriented scale"
    - The learning rate in front of the weight decay term being the square of the learning rate scaling the gradient ($\xi^2 / \hat{v}_t$ vs $\xi/\sqrt{\hat{v}_t}$)
    - The multiplier $M_2(g_t)$ in front of the weight decay term.
    - The decay scheme for the global step size ($d_t$)
    - The additional decay scheme for the weight decay term ($c_t$)
    - Computing $\hat{v}_t$ across model parameters instead of using running averages.

2. The derivation of Amos is a series of approximations (some very crude) and heuristics/intuition. I could put some caveats on almost all the steps of the derivation. While the authors clearly state that this is a heuristic derivation, at some point we have to ask ourselves whether such a derivation is helpful or just obfuscates. Some steps that I found particularly problematic:
    - Equation (6) suggests a particular _ratio_ of $\alpha_t^2$ to $\rho_t$. It is completely unclear to me how that should motivate $\rho_t \propto M_2(g_t)^2 / \mathbf{E}[M_2(g_t)^2]$ and $\alpha_t \propto 1/\sqrt{\mathbf{E}[M_2[g_t]^2]]}$. This choice satisfies Eq. (6), but so does an infinite number of other choices.
    - Assumption 1: This assumes $\mathbf{E}[\Vert g_t\Vert^2] \leq C \cdot \Vert \mathbf{E}[g_t]\Vert^2$, i.e., the variance of a stochastic gradient needs to go to zero as the true (expected) gradient goes to zero. That is the so-called interpolation regime: every data point is fit perfectly. This seems unlikely to be fulfilled in LLM pretraining tasks.
    - Interpretation of $\xi$. Sorry, but this is almost absurd. First, if we assume the per-example gradients to come from a zero-mean distribution this would imply $\mathbf{E}[g_t] = 0$, in direct contradiction to the Equation below. Second, why should  $$\left(\frac{1}{N} \sum_{i=1}^N x_i \right)^2 = \frac{1}{N^2}\sum_{i=1}^N x_i^2$$ hold? That is simply not true and is not a meaningful approximation in any way. Third, the law of large numbers is completely irrelevant here.

3. The experiments focus entirely on Transformer architectures. While this is certainly a model class of great practical importance, I would find it highly desirable to check the extent to which these findings transfer to other families of architectures and data modalities. Given the resources that went into the experiments involving Transformers, it would be "cheap" to throw in a ResNet on CIFAR-10/CIFAR-100/ImageNet. If Amos performs amicably there - great. If not, this would alert us of the fact that some of the proposed heuristics are specific to Transformers (which would be perfectly fine).

**Summary Of The Paper:**

This paper proposes Amos, an adaptive optimization method for deep learning. It combines various aspects, most notably an adaptive weight decay term and a step size involving an expected scale of model parameters, informed by structural knowledge about the model being trained. It is evaluated in extensive experiments on pretraining Transfomer models.

**Summary Of The Review:**

I list several scientific shortcomings above, which are think are pretty fundamental. In light of that, the good experimental results for the proposed method on their own do not warrant acceptance in my opinion.

---

> ### Author Response · Authors · 2022-11-19
> **We have rewritten the derivation section and added ImageNet experiments; Reply with clarification and welcome discussion**
>
> We appreciate the reviewer comments, and have revised the paper significantly (clearer derivation, ImageNet experiment, etc.). Specific answers are provided below.
>
> * _the method combines a large number of aspects without proper ablations_
>
> To clarify, the components in Amos (which we divide into three, to align with AdamW: a gradient descent part, a decoupled weight decay part, and decay schedule) are not “largely independent”. For example, the scale of trained weights depends on both the learning-rate of the gradient descent and strength of the weight decay. In order to make the weights converge to specified scale “eta”, the learning-rate and the weight decay strength should satisfy Equation 5 in our revised paper (we did try settings in our preliminary experiments that do not satisfy Equation 5, as described below). Furthermore, it is known that a decaying schedule for the learning-rate is necessary for better performance, which implies that the strength of weight decay should be decreased accordingly due to our theory (in the new ResNet50 on ImageNet experiments, we also tested a setting with constant weight decay strength). Moreover, we have already conducted several ablative tests to illustrate how the Amos optimizer works, as discussed below.
>
> The decaying schedule is indeed crucial for the performance, for which we have tried some different settings in our preliminary experiments, as described below. Our derivation of the decaying schedule is heuristics-heavy, but it is not arbitrarily introduced; we welcome the reviewers to discuss with us about the assumptions and possible alternatives, some settings we might have already tested but some may not.
>
> * _The factored learning rate involving the "model-oriented scale"_
>
> We have modified the factor “eta” in the RPE model and BERT-large (the Amos-*Scale curve in Figure 1c and Figure 2a, Page 7-8) to show that the factor can actually control the scale of trained variables and affect performance. And we have run a grid search for the global learning rate (Figure 5 in Appendix A.5, Page 18-19).
>
> * _The learning rate in front of the weight decay term being the square of the learning rate scaling the gradient_
>
> This actually makes a lot of sense. As we described in Page 5 of the revised paper, $g_t$’s upon different $z_t$’s are randomly noisy, so they will cancel out when being averaged; by Assumption 1, the magnitude of the averaged gradient is about $\xi$ of the magnitude of $g_t$. On the other hand, $\theta_t$ does not depend on $z_t$ and it changes slowly, so the weight decay is easier to accumulate. An extra $\xi$ factor will balance weight decay with the gradient descent. Specifically for pre-training Transformers, we use $\xi=0.01$; so changing the weight decay strength from $\xi^2$ to $\xi$ (i.e. increasing it 100 times) will make the weights decaying too fast and largely hurt the performance (we have actually tried this in preliminary experiments).
>
> * _The multiplier $M_2(g_t)$ in front of the weight decay term_
>
> This term makes the weight decay adaptive, and it is a direct implication from Equation 5 in the revised paper. We have tested a setting with constant weight decay strength in the new ResNet50 on ImageNet experiments.
>
> * _The decay scheme for the global step size $d_t$, The additional decay scheme for the weight decay term $c_t$_
>
> Our theoretical derivation has suggested the asymptotic behavior of $c_t$ and $d_t$, i.e. when $t$ is large, $c_t$ should decay in order of $b_t^{-1/2}$ and $d_t$ in order of $b_t^{-1}$. We have tried different settings in our preliminary experiments, e.g. $d_t$ decays in order of $b_t^{-1/2}$ instead, but the convergence was not as fast. We have also tried different settings with the constants $p$, $q$: It seems that $q\propto\sqrt{\eta}$ is necessary for the trained variables to converge to scale $\approx\eta$; we have tried $q\propto\eta$ or $q$ does not depend on $\eta$, but didn’t work. After $c_t$ and $d_t$ are fixed, they seem to generalize to other models and tasks, e.g. in our new experiments with ResNet50 on ImageNet.
>
> We do not present these preliminary experiments in the paper, because without the constraints from our theory, the search space for decay factors is vast and there is no obvious target to be compared with (except the SOTA setting of AdamW with linear learning-rate decay). Nonetheless, the good performance of Amos on pre-training as presented in this paper is indeed not trivial, as previous works have tried a large amount of different learning-rate schedules (e.g. Appendix D.6 in Kaplan et al. Scaling Laws for Neural Language Models).
>
> * _Computing_ $\hat{v}_t$ _across model parameters instead of using running averages_
>
> This is corresponding to our memory reduction strategy, for which we have compared different settings (Figure 7 in Appendix A.8, Page 19-20).

---

> > ### Author Response · Authors · 2022-11-19
> > **Reply to concerns about scientific quality**
> >
> > We have revised the paper and reorganized the heuristic derivation section to make it clearer based on your feedback, and also added an experiment on ResNet50 (Appendix A.9). That said, we believe the heuristic derivation does not reduce the scientific quality of this work.
> >
> > Given the complexity of data and models we encounter in reality, it is almost inevitable for a working theory to include some meaningful logical leaps and guessing. Specific to this work, we raised the problem of how to mix gradient descent and weight decay in a balanced way, in order for an optimizer to lead model weights to converge to a specific scale. This is intuitively achievable but no doubt difficult; and conventional theories, e.g. regret bounds, are unlikely to provide insights for this problem. Heuristic derivations based on intuitions and assumptions are common in physics, which do not necessarily degrade the scientific value; a good example is that Max Planck invented the concept of quantized energy when trying to heuristically fit two empirical formulas of black-body radiation. Specific to this work, we very much welcome the reviewers to discuss our intuitions and heuristics - some logical gaps we are aware of, some maybe not.

---

> > ### Author Response · Authors · 2022-11-19
> > **Reply to clarify the steps in our derivation**
> >
> > We have rewritten the section of derivation, and hope the new version can help resolve several potential misunderstandings. We welcome the reviewers to discuss with us about the intuitions, assumptions, logical structure of the derivation, and raise their questions. The specific questions raised here are discussed below.
> >
> > * _This choice satisfies Eq. (5), but so does an infinite number of other choices_
> >
> > There are not as many other choices under the requirements that $\alpha_t$ does not depend on $z_t$ and $\mathbb{E}[\rho_t]$ does not depend on $g_t$. For example, one might think that $\rho_t\propto g_t$ and $\alpha_t\propto 1/g_t$ can also satisfy the equation; but this violates the requirement that $\alpha_t$ does not depend on $z_t$. We have formalized this step as Lemma 4.2 in the revised paper.
> >
> > * _Assumption 1: This assumes the variance of a stochastic gradient needs to go to zero as the true (expected) gradient goes to zero. That is the so-called interpolation regime: every data point is fit perfectly. This seems unlikely to be fulfilled in LLM pretraining tasks._
> >
> > Assumption 1 indeed shares a similar intuition as the interpolation regime, which we have cited; but the focus of Assumption 1 is different, and this assumption is verified by our experiments. We clarified this in Footnote 2 (Page 6 in the revised paper): Assumption 1 only requires that $M_2(g_t)$ decreases as fast as $\mathbb{E}[g_t]$, which is empirically verified in Section A.6. Whether $M_2(g_t)$ actually converges to 0 is not guaranteed (because the training may stop early, or $\mathbb{E}[g_t]$ not get to exactly 0 due to L2-regularization, etc.) and not used in our theory. On the other hand, $M_2(g_t)$ is always large compared to $\mathbb{E}[g_t]$, because $\xi$ is a small value.
> >
> > * _Interpretation of $\xi$_
> >
> > First, for finite N samples drawn i.i.d. from a zero-mean distribution, the empirical average of these samples is not exactly 0 (so there is no direct contradiction; also, a set of samples can be viewed as drawn from two different distributions, one with mean 0 and one not, just like what we do in hypothesis testing, and no contradiction in there either). In fact, the expectation of the square of the empirical average is 1/N of the variance of the distribution, which is a conclusion of the Law of Large Numbers, and is exactly what we have here: The numerator is the square of the empirical average, and the denominator is an estimation of the variance of the distribution. By taking the quadratic mean $M_2(\bullet)$, we are also averaging across model parameters here, so the numerator is actually the **expectation** of the square of the empirical average, due to the Law of Large Numbers again.
> >
> > To directly answer the question of the reviewer: why should
> >
> > $$\left(\frac{1}{N}\sum_{i=1}^{N}{x_i}\right)^2=\frac{1}{N^2}\sum_{i=1}^{N}x_i^2$$
> >
> > hold? Because the cross terms all have expectation 0, so their sum will cancel out on average. This is actually how we prove the Law of Large Numbers.

---

> > ### Author Response · Authors · 2022-11-19
> > **Reply to "throw Amos in a ResNet on CIFAR-10/CIFAR-100/ImageNet"**
> >
> > We have added experiments with ResNet50 on ImageNet, and Amos works quite well out-of-the-box (Section A.9). Thanks for the advice.

---

### Official Review · Reviewer_Ebn1 · 2022-11-01

**Confidence:** 3
**Correctness:** 2
**Technical Novelty And Significance:** 3
**Empirical Novelty And Significance:** 3
**Recommendation:** 5

**Clarity, Quality, Novelty And Reproducibility:**

* Regarding novelty (see my notes from before): I think this is a really nice and promising idea, and it is well embedded in recent and relevant literature.

* Regarding quality (see my notes from before): I think the execution of the idea is heuristic-heavy, and contains a substantial degree of arbitrariness and complexity that is not justified in a compelling way. Experiments are very promising, but in my opinion do not make up for this.

* I had several clarity issues reading the paper, related to notation, formulation, experiments, paper structure and general model clarity:
  * The explanation for obtaining eta is unclear to me ("match input/output range", "not dominate"). Concept of "range" and its importance is not introduced. It seems that it would depend on weight distribution as well?
  * Model-oriented scale: overloading of variable W is confusing.
  * Notation: the lack of distinction between scalar+global variables (eta, c) and per-partition scalars hinders clarity. Indexing partitions would help
  * Gamma: It is unclear how this component "makes trained weights empirically converge to eta_tilde". If eta_tilde is the norm for the full model, how does gamma achieve this without using that global information?
  * weight decay: Could be expressed in a clearer way. Why is it affected by the decay multiplier "d", while it also has its own decay factor "c"? The explanation in 4.2 introducing L2 doesn't seem to address this
  * Section 4.2 aims to provide a "heuristic derivation", but it is very difficult to follow since it is not lemma-oriented. At many points, it is not clear what are we expecting to see, and how exactly the discussed equations are related to Amos. E.g. after equation 5: "now recall that we are given eta such that...": this was claimed, but I don't see in the paper where this is satisfied. In general, I found the section very confusing as presented. My suggestion would be to resort to a lemma-oriented structure, and whenever the derivations aren't possible or don't lead to watertight conclusions (e.g.  inequalities, broad assumptions), provide experiments that support the ideas presented (e.g. tight bounds), as done in Appendix A.5.
  * Experimental benchmarks: What is the difference between pre-training and regular training when comparing the observed results? If we are expected to train afterwards, wouldn't the final result (including the training after pre-training) be the actual target to compare? Note that recent results on "critical learning periods" (e.g. Achille et al 19). show that issues with pre-training can affect final performance.

* Regarding reproducibility: The paper contains a clear description of the steps to be computed. The experiments are based on well-known, public architectures and benchmarks. Quantitative results are provided, although without error bars. Resources required are high but not unreachable. For the missing details, the authors pledge to provide an open-source (JAX) implementation, upon publication. If that is the case and assuming no lucky seeds, reproducibility issues are expected to be minimal.

**Strength And Weaknesses:**

The idea of using model information to replace the hyperparameters is great, we have good examples of similar strategies from the weight initialization literature (Xavier, He) and extending this to adaptive optimizers sounds exciting. Furthermore, this combines also very nicely with the interplay between learning rate and decoupled weight decay, which has been recently proven useful to solve issues with first-order adaptive DL optimizers like Adam. Finally, it also brings better performance using less parameters in a block-wise regularized fashion.
Combining these ideas is a very promising and well-grounded line of work.

Still, the question of why specifically this model, is not fully clear to me, for the following reasons:
* Compared to its counterparts, the model introduces increased complexity, but the motivation (section 1) lists reducing complexity as a goal. This is expected to be achieved through the "guidance for hyperparameter tuning", but I'm not convinced of such guidance (see next points).
* There is a clear effort to justify the choice for heuristics and hyperparametrizations, but the efforts lack clarity both in substance and presentation (see notes on clarity) and the claim that they are "theoretically supported" is in many cases not true: many crucial aspects are resolved through trial-and-error (see e.g. end of page 6 and Appendix 7). In many cases, I feel like an ablation study showing the contribution of the added components in an empirical way would be clearer and more compelling.
* Even the more analytical hyperparameters don't give the impression of being "easier" to find. E.g. for "eta", Appendix A.3 shows that many factors must be taken into account in a non-automated way, and new architectures may require careful tuning. What if we e.g. have batchnorm instead of layernorm? How is the concept of "range" characterized? Will it hold under non-stationary, noisy and/or sparse gradients?
* Regarding stability: While very promising, the experiments aren't comprehensive enough to convince that the proposed settings (theoretical or empirical) are not highly specific to the results reported, due to the amount of hyperparameters and tuning involved. One thing that could help with this is to analyze the loss landscape stability (see AdamW paper), and/or extend experiments to other tasks and architectures.
* Regarding experiments, it is unclear how many settings were tried before finding the reported Amos hyperparametrizations. This makes difficult to compare the different optimizers in terms of their hyperparametrization budgets. Furthermore, error bars are not provided: they would be welcome (together with a more publication-friendly plotting mechanism), although the differences between Amos and the rest are very significant.

A couple of thoughts regarding weight decay:
* Please correct footnote in page 2: Loschilov&Hutter precisely mentions that L2 and weight decay are not equivalent for adaptive optimizers (beyond empirical success), the footnote seems to operate on the opposite idea.
* We saw that decoupled weight decay helps overcoming convergence issues with Adam. Would re-introducing adaptive weight decay also re-introduce some of those issues?


**Summary Of The Paper:**

The paper introduces Amos, a first-order DL optimizer with adaptive learning rate and decay. The proposed contributions are:

* Outperforming AdamW for pre-training language models
* Providing guidance for hyperparameter tuning
* Reducing memory usage
* Allowing continuous training and resuming from checkpoints

The proposed optimizer leverages model-specific information, by partitioning the model parameters and adding a per-partition norm constraint. This norm constraint, together with the gradients, is used to adapt the learning rate and weight decay.
The parameter update involves a series of newly introduced hyperparameters with different functions, and the paper provides a series of heuristic derivations and experiments to set their values.

In experiments pre-training current architectures for NLP tasks, Amos is shown to outperform AdamW with 2019 settings in terms of speed, performance and memory usage.

**Summary Of The Review:**

My opinion is that this contribution clearly deserves attention from the community, but it needs more work: It presents an optimizer of increased complexity, involving a substantial amount (potentially arbitrary) heuristics. Experiments are very promising, but there are methodological concerns.
The paper would greatly benefit from improvements in clarity, both in formulation and presentation, especially given the increased amount of details needed to understand and tune this optimizer.

For those reasons I'm inclined to marginally reject, but I thank the authors for their contribution and encourage them to address/answer some of my points above in order to reconsider my evaluation.

---

> ### Author Response · Authors · 2022-11-19
> **We have significantly revised the paper with clearer derivation and ImageNet experiments**
>
> We appreciate the reviewer comments, and have revised the paper significantly (clearer derivation, ImageNet experiments, etc.).
>
> * _Compared to its counterparts, the model introduces increased complexity_
>
> From a high point of view, Amos has the same main components as AdamW, i.e. there are three moving parts: a gradient descent part, a decoupled weight decay part, and a learning-rate decay. And the number of hyper-parameters that require tuning is actually less: We do not need to set a weight decay strength or max training step, as required by AdamW with decayed learning-rate. For Amos, only the global learning-rate is tuned, and even for this we have provided an intuitive estimate that can usually serve as a good start for tuning.
>
> It is true that each component of Amos has been further decomposed into several factors, for example the initial learning-rate is decomposed into a global learning-rate and a model-oriented scale factor. But this is not necessarily a weakness, because these factors combined with our theory provide more interpretable details to the moving parts, and  especially the number of tunable hyper-parameters has decreased.
>
> That said, there is the concern that the proposed mechanism in Amos might only work for pre-training certain types of language models as we presented in the paper, but may not generalize to other models and tasks. That is because our theoretical derivation is heuristics-heavy and based on several intuitions that may not generalize to all situations. To answer this concern, we have rewritten the heuristic derivation section of the paper, with all the intuitions and assumptions explicitly clarified, and we added new experiments with the ResNet50 model on ImageNet.
>
> * _clarity of theoretical derivation; ablation of added components_
>
> Regarding the theoretical derivation, we have rewritten the section to put more focus on issues raised here and by other reviewers, and hope the new version can help resolve several potential misunderstandings.
>
> Our theory added more detailed explanations to existing moving parts, and we have conducted several ablation tests to justify our explanation: (1) We have modified the factor “eta” in the RPE model and BERT-large to show that the factor can actually control the scale of trained variables and affect performance; (2) We have run a grid search for the global learning rate; (3) We have compared different memory reduction strategies.
>
> Specific to ablation of the decay factors: The form of the decay factors $c_t$ and $d_t$ is indeed crucial for the good performance on pre-training Transformer variants. Our theoretical derivation has suggested the asymptotic behavior of $c_t$ and $d_t$, i.e. when $t$ is large, $c_t$ should decay in order of $b_t^{-1/2}$ and $d_t$ in order of $b_t^{-1}$. We have tried different settings in our preliminary experiments, e.g. $d_t$ decays in order of $b_t^{-1/2}$ instead, but the convergence was not as fast. We have also tried different settings with the constants $p$, $q$: It seems that $q\propto\sqrt{\eta}$ is necessary for the trained variables to converge to scale $\approx\eta$; we have tried $q\propto\eta$ or $q$ does not depend on $\eta$, but didn’t work. After $c_t$ and $d_t$ are fixed, they seem to generalize to other models and tasks, e.g. in our new experiments with ResNet50 on ImageNet.
>
> * _"eta" is not "easier" to find_
>
> While we agree that there is not a unique value of eta that is always appropriate, we believe our choices are theoretically justified. Moreover, eta is defined at the component level and we have covered many components (normalization layer,  nonlinear activation, Attention, MLP, CNN, RNN, max-pooling, etc.). Thus, it would be possible to automate the calculation with the help of modeling libraries, similar to initialization, for an architecture that is composed of these components. One might always be tempted to “tune” the value of eta in some specific way, but in our experience, the correctly calculated eta almost always works the best.
>
> To answer the specific questions: batchnorm can be treated the same as layernorm, see our experiments with ResNet50 in the revised paper; the concept of “range” should be defined as the standard deviation of the hidden tensors, which we have clarified in our revision.
>
> * _extend experiments to other tasks and architectures_
>
> We have added experiments with ResNet50 on ImageNet, and Amos works quite well out-of-the-box.
>
> * _hyper-parametrization budgets_
>
> As curious researchers, we have tried a lot of different settings in our preliminary experiments, some even against our theory or intuition, just to get a better understanding of how each part works. As an Amos user, one only wants to tune the global learning rate $\xi$ and the running average rate $\beta$, for which we have provided guidance for tuning. We have run the training experiments presented in the paper several times; the learning curve is stable and does not depend on random seeds.

---

> > ### Author Response · Authors · 2022-11-19
> > **Reply to a couple of thoughts regarding weight decay**
> >
> > * _footnote in page 2_
> >
> > Thanks for pointing this out. We have improved the footnote to clarify. Our understanding is that after Loschilov&Hutter’s work, newly proposed adaptive optimizers mostly include an explicit decoupled weight decay term (e.g. AdaFactor), which we regard as the more appropriate form of L2 regularization. We still want to use the term “L2 regularization” in order to inherit the intuition, and to compare it with L2-regularized SGD.
> >
> > * _We saw that decoupled weight decay helps overcoming convergence issues with Adam. Would re-introducing adaptive weight decay also re-introduce some of those issues?_
> >
> > This is a good point. In Equation 3 of our revised paper, although the L2 strength $\rho_t$ is adaptive (i.e. depending on $g_t$), we have required that $\mathbb{E}[\rho_t]$ does not depend on $g_t$, which is achieved by normalizing $\rho_t$ with $\mathbb{E}[g_t^2]$. So on average the L2 strength is not affected by  the typical gradient magnitude on each variable. We made this clear in the revision.

---

> > ### Author Response · Authors · 2022-11-19
> > **Reply to clarity issues**
> >
> > * _The explanation for obtaining eta is unclear to me ("match input/output range", "not dominate"). Concept of "range" and its importance is not introduced. It seems that it would depend on weight distribution as well?_
> >
> > “range” should be understood as the standard deviation of the hidden tensors. Although not completely rigorous, assuming the distribution to be Gaussian is a good approximation most of the time. And yes, we intend to assume that the entries of weights are Gaussian distributed as well. This is made clear in the revision.
> >
> > * _Model-oriented scale: overloading of variable W is confusing._
> >
> > Fixed.
> >
> > * _Notation: the lack of distinction between scalar+global variables (eta, c) and per-partition scalars hinders clarity. Indexing partitions would help_
> >
> > Scalars and vectors are mixed in our equations due to the memory reduction mechanism, which is somehow the nature of this work; we have added more explanations in the revised paper. On the other hand, we would still like to omit partition indices for simplicity, but made clearer which are per-partition scalars and which are global.
> >
> > * _Gamma: It is unclear how this component "makes trained weights empirically converge to eta_tilde". If eta_tilde is the norm for the full model, how does gamma achieve this without using that global information?_
> >
> > This is a good question. The scale of trained weights depends on both the learning-rate and L2 strength. Their relation is governed by Equation 5 in the revised paper. In other words, even if the L2 strength is the same, a smaller learning-rate will result in a smaller scale of trained weights. In our preliminary experiments, it seems better to set the L2 strength independent of the scale of each variable (which may be related to the fact that the L2 strength coincides with the decay speed, so an L2 strength independent of the scale of each variable might lead all variables converge in the same speed), which also complies with the intuition that L2 regularization is a global process applied to the whole model, rather than each specific variable. Conversely, with alternative settings where L2 strength depends on eta_tilde, even if we can get the trained weights converge to the specified scale, our preliminary experiments suggest that the training performance is usually worse.
> >
> > * _extra decay factor "c"_
> >
> > The extra decay factor “c” ensures that $\rho_t$ decays faster than $\alpha_t$, which is required by our theory, as made clearer in our revision. This effect is indeed observed in practice: for example, with AdamW the weight decay strength is set to a constant, so the weight decay will become relatively stronger as the learning-rate decays to 0, which usually results in the phenomenon that the scale of trained weights decreases close to the end of training.
> >
> > * _heuristic derivation is very difficult to follow since it is not lemma-oriented_
> >
> > Section 4 is rewritten in our revised paper using lemmas as suggested. Thanks for the advice. In particular, the logic chain is made clearer: the equation $M_2(\theta^\ast)\approx\eta$ is used to derive a necessary condition (i.e. Equation 5 in the revised paper) for the trained weights to converge to scale $\eta$, rather than proving that the necessary condition actually leads to $M_2(\theta^\ast)\approx\eta$.
> >
> > * _What is the difference between pre-training and regular training when comparing the observed results?_
> >
> > The difference in pre-training is that the training process is longer so the decaying schedule is crucial; and the training data is large so no need to tune the L2 regularization. Amos is likely to perform better under these two conditions. On the other hand, we have also included the experiments of finetuning on MNLI (Section A.7), regular training LSTM on PTB (Section 5.2), and ResNet50 on ImageNet (Section A.9).

---

### Author Response · Authors · 2022-11-19
**Top Level Response**

We appreciate the reviewer’s constructive feedback and have made the following main changes in the revision of the paper:

1. We significantly revised the theoretical derivation (Section 4) based on the reviewer feedbacks.

2. We have added additional experiments for ResNet50 on ImageNet, showing that Amos performs well “out-of-the-box” to a different domain (Appendix A.9).

Individual reviewer comments are addressed below.

Regarding the correctness and clarity concerns raised by all the reviewers, we would like to emphasize the complexity in nature of the problem that we are trying to solve:

**How to mix gradient descent and weight decay in a balanced way, in order for an optimizer to lead model weights to converge to a specific scale?**

This goal is intuitively achievable but no doubt difficult; and conventional theories are unlikely to provide insights for the solution.

Our proposed optimizer, Amos, works very well in practice; and although the theoretical derivation of Amos is largely based on intuitions and heuristics, we believe it has the value in guiding the research of optimization, and the intuitions and heuristics are worth discussing. We very much welcome, invite and encourage all the readers to discuss with us, about any questions or less clear aspects in our theory.

---

> ### Author Response · Authors · 2022-11-30
> **Share Tensorboard Logs of Pre-training Experiments**
>
> Dear AC and Reviewers:
>
> As an additional note, we would like to share the Tensorboard logs of pre-training experiments (i.e. Figure 1 and Figure 2) in the paper as below.
>
> * Figure 1: https://tensorboard.dev/experiment/taN9wX92T9WvmrMjPHz7tQ/
> * Figure 2, BERT and RoPE: https://tensorboard.dev/experiment/wOsggWvZTx2qSqvOXWbBDA/
> * Figure 2, T5X: https://tensorboard.dev/experiment/FMe95N5vQ56aBViLrx7WGg/
>
> These logs also provide more detailed data than presented in the paper, including the scales of all trained variables in the models.
> Hopefully they could help for closer examination and easier comparison of the experiment results.
>
> And we would be happy to share more experiments, including the preliminary ones described in our comments, under the request of the reviewers.
> Again, any questions or discussions are welcome.
>
> Best,

---

### Decision · Program_Chairs · 2023-01-20

**Decision:**

Reject

**Justification For Why Not Higher Score:**

Well over 100 optimization algorithms for deep learning have been published over the past 7 years. A paper proposing another one should have to clear a high bar for novelty and performance.

**Justification For Why Not Lower Score:**

N/A

**Metareview: Summary, Strengths And Weaknesses:**

While all three reviewers noted some strengths of this paper, they agree in their evaluation that this paper is not sufficiently convincing to be accepted. The authors have responded in great detail, but these do not resolve all the issues raised by the reviewers.